# Efficiently Learning Significant Fourier Feature Pairs for Statistical Independence Testing

**Yixin Ren[1], Yewei Xia[1,4], Hao Zhang[3,\*], Jihong Guan[2,\*], Shuigeng Zhou[1,\*]**

[1]Shanghai Key Lab of Intelligent Information Processing, and School of
Computer Science, Fudan University, Shanghai, China
[2]Department of Computer Science and Technology, Tongji University, Shanghai, China
[3]SIAT, Chinese Academy of Sciences, Shenzhen, China
[4]Machine Learning Department, MBZUAI, Abu Dhabi, UAE
{yxren21, ywxia23}@m.fudan.edu.cn, h.zhang10@siat.ac.cn
jhguan@tongji.edu.cn, sgzhou@fudan.edu.cn

## Abstract

We propose a novel method to efficiently learn significant Fourier feature pairs for maximizing the power of Hilbert-Schmidt Independence Criterion (HSIC) based independence tests. We first reinterpret HSIC in the frequency domain, which reveals its limited discriminative power due to the inability to adapt to specific frequency-domain features under the current inflexible configuration. To remedy this shortcoming, we introduce a module of learnable Fourier features, thereby developing a new criterion. We then derive a finite sample estimate of the test power by modeling the behavior of the criterion, thus formulating an optimization objective for significant Fourier feature pairs learning. We show that this optimization objective can be computed in linear time (with respect to the sample size $n$), which ensures fast independence tests. We also prove the convergence property of the optimization objective and establish the consistency of the independence tests. Extensive empirical evaluation on both synthetic and real datasets validates our method's superiority in effectiveness and efficiency, particularly in handling high-dimensional data and dealing with large-scale scenarios.

## 1 Introduction

Testing for independence is a crucial and challenging task in machine learning and statistics, with wide-range applications in causal inference [16, 31], feature selection [6] and deep learning [23, 42]. Its primary objective is to determine whether two random variables, $X$ and $Y$ are independent, based on the observations of the underlying joint distribution $\mathbb{P}_{XY}$. While traditional independence tests, such as Pearson's correlation coefficient [9] and Kendall's $\tau$, can only detect monotonic relationships between low-dimensional variables, more modern tests [26, 43, 7, 25, 27, 35, 19, 20] aim to deal with complex non-linear interactions in much more challenging higher-dimensional space [45, 29].

One class of nonlinear dependence measures [3, 15] aims to capture distributional characteristics using kernel embeddings [13], primarily derived from the cross-covariance operators in the reproducing kernel Hilbert space (RKHS). Among them, Hilbert-Schmidt Independence Criterion (HSIC) [14] is the most popular one. It utilizes the squared Hilbert-Schmidt norm to detect dependence and exhibits outstanding performance across various data contexts by choosing suitable kernels. On the other hand, some other fundamental nonlinear dependence measures employ characteristic functions

---

*Corresponding authors.

38th Conference on Neural Information Processing Systems (NeurIPS 2024).

to detect the smoothed discrepancy between the joint distribution and the product of marginals. By employing appropriate characteristic functions, the statistic [39, 40] computes the covariance between distances of variable pairs. It has been demonstrated that these distance-based methods are equivalent to HSIC with specific kernels [33]. However, all these measures suffer from the drawback of requiring quadratic time (w.r.t. the sample size $n$) to compute the feature covariance and necessitating fixed kernel or distance functions, rendering them impractical on large-scale datasets due to the unaffordable time cost and lacking flexibility in handling complex scenarios.

To address these challenges, a multitude of works grounded on these measures have emerged. Upon HSIC, [44] proposes some linear-time tests including a block-averaged statistic, a statistic with Nyström approximation, and one with finite-dimensional feature mappings using random Fourier features (RFF) [28]. For convenience, these tests are referred to as BHSIC, NyHSIC, and FHSIC, respectively. FHSIC and NyHSIC are observed to have a considerable advantage over BHSIC. However, a remaining drawback of these methods is that the features are not learnable. Therefore, these methods lack enough adaptability to complex settings, thus leading to performance degradation.

In addition to time efficiency, another research direction [1, 30] aims to make independence tests adaptive to better capturing distributional distinctions. These methods either select/combine appropriate kernels from a predefined set or learn parameterized kernels. Nonetheless, their criteria still inherit the quadratic time complexity of HSIC, thus cannot be readily applied to large-scale data.

Furthermore, some approaches [17, 32] try to address both challenges simultaneously. For instance, HSICAgg [32] suggests combining several kernels from a predefined set (e.g. kernels with different preset bandwidths) and aggregating the test results for improving performance. Additionally, an incomplete $U$-statistic of HSIC is proposed to ensure computational efficiency. Nevertheless, selecting from a predefined set of kernels imposes limitations on flexibility, and in cases where scaling optimization is required on each dimension, the number of kernel pairs escalates exponentially. Also, NFSIC [18] proposes to combine a time-efficient technique called analytic kernel embeddings [8, 17] and learn the important local distributional features. However, its learning objective is merely a lower bound of test power and demands a substantial number of samples to ensure accuracy.

In this paper, we propose a novel test method that flexibly learns distributional features while maintaining high efficiency. We first reinterpret HSIC from a frequency-domain perspective, then we point out its potential shortcomings with an elaborate example and indicate corresponding improvement directions. Finally an central optimization objective is derived by directly modeling test power, which can be computed in linear time while maximizing the test performance. Comparing with [30] that also addresses the kernel learning problem in independence testing with a time/space complexity of $\mathcal{O}(n^2)$, our criteria for learning are designed to have a complexity of $\mathcal{O}(n)$ for both space and time. Consequently, the whole test framework can efficiently handle large-scale data.

**Contributions.** In summary, the contributions of the work are as follows: 1) We propose a novel approach that efficiently learns significant Fourier feature pairs for maximizing the power of HSIC-based independence tests. 2) We design an optimization objective that can be computed in linear time, which is derived by directly modeling test power. 3) We theoretically establish the non-asymptotic convergence property of the optimization objective and demonstrate the consistency of our method. 4) We conduct extensive experiments on both synthetic and real data, showcasing its superiority in effectiveness and efficiency in handling high-dimensional data (e.g. image data) and addressing large-scale scenarios.

**Outline.** The rest of the paper is organized as follows: Sec. 2 reviews HSIC-based statistical independence tests. Sec. 3 reinterprets HSIC from a frequency-domain perspective, and explain its potential shortcomings with an elaborate example and indicate corresponding improvement directions. Sec. 4 designs an optimization objective by directly modeling test power, which can be computed in linear time. Sec. 5 presents the theoretical analysis and Sec. 6 evaluates the performance of the proposed method on synthetic and real dataset. We conclude the paper in Sec. 7.

## 2    Preliminaries and Notations

We begin by introducing notions and reviewing the hypothesis testing framework for independence tests. Let $\mathcal{X} \times \mathcal{Y}$ be separable metric space, typically $\mathbb{R}^{d_x} \times \mathbb{R}^{d_y}$. $\mathbb{P}_{XY}$ denotes a Borel probability measure defined on $\mathcal{X} \times \mathcal{Y}$, while $\mathbb{P}_X$ and $\mathbb{P}_Y$ denote the respective marginal distributions. Given $n$

independent and identically distributed (i.i.d) samples $Z := (X, Y) = \{(x_i, y_i)\}_{i=1}^n$ with distribution $\mathbb{P}_{XY}$, we aim to test whether $X, Y$ are independent (i.e., $X \perp\!\!\!\perp Y$). This corresponds to a hypothesis testing problem formulated as $\mathcal{H}_0 : \mathbb{P}_{XY} = \mathbb{P}_X \mathbb{P}_Y$ versus $\mathcal{H}_1 : \mathbb{P}_{XY} \neq \mathbb{P}_X \mathbb{P}_Y$.

The testing procedure is as follows: First, define the statistic $\rho$ and calculate its estimated value using the samples. Then, choose a significance level $\alpha$ (typically set to $0.05$), which represents the probability that the sampling of $\rho$ under $\mathcal{H}_0$ is at least as extreme as the observed value. Finally, the null hypothesis $\mathcal{H}_0$ is rejected if the $p$-value is not greater than $\alpha$.

Two types of errors may occur in this procedure. Type I error occurs when $\mathcal{H}_0$ is falsely rejected, while Type II error happens when $\mathcal{H}_0$ is incorrect but not rejected. A good test [43] needs to control Type I error within $\alpha$ while maximizing the testing power ($1 - $Type II error rate).

For independence tests, a commonly used statistic is HSIC, defined as follows:

**Definition 1.** *[14]. Let $\mathcal{F}$ be an RKHS with kernel $k : \mathcal{X} \times \mathcal{X} \mapsto \mathbb{R}$ and $\mathcal{G}$ be a second RKHS on $\mathcal{Y}$ with kernel $l : \mathcal{Y} \times \mathcal{Y} \mapsto \mathbb{R}$, the HSIC between $X$ and $Y$, denoted as HSIC$(X, Y)$ is defined as*

$$\mathbf{E}\big[k(X, X')l(Y, Y')\big] + \mathbf{E}\big[k(X, X')\big]\mathbf{E}\big[l(Y, Y')\big] - 2\mathbf{E}_{X'Y'}\big[\mathbf{E}_X k(X, X')\mathbf{E}_Y l(Y, Y')\big], \quad (1)$$

*where $(X', Y')$ is a independent copy of $(X, Y)$. An estimator of HSIC$(X, Y)$ is given by*

$$HSIC_b(Z) := \frac{1}{n^2}\sum_{i,j} k_{ij}l_{ij} + \frac{1}{n^4}\sum_{i,j,q,r} k_{ij}l_{qr} - 2\frac{1}{n^3}\sum_{i,j,q} k_{ij}l_{iq} = \frac{1}{n^2}Tr(\mathbf{KHLH}), \quad (2)$$

*where $k_{ij} := k(x_i, x_j)$, $l_{ij} := l(y_i, y_j)$ are the entries of the $n \times n$ kernel matrices $\mathbf{K}$, $\mathbf{L}$ respectively, $\mathbf{H} = \mathbf{I} - \frac{1}{n}\mathbf{1}\mathbf{1}^T$ is the centering matrix and $\mathbf{1}$ is a vector of ones.*

## 3 Revisiting HSIC from Frequency Domain Perspective

We denote $\mathcal{F}$ as the Fourier transform, and $\mathcal{F}^{-1}$ as its inverse. When the kernels $k, l$ are translation-invariant, i.e., there exist functions $\psi, \psi_k, \psi_l$ such that for all $(x, x') \in \mathcal{X} \times \mathcal{X}$ and $(y, y') \in \mathcal{Y} \times \mathcal{Y}$,

$$\psi(x - x', y - y') = \psi_k(x - x')\psi_l(y - y') = k(x, x')l(y, y'). \quad (3)$$

Then, according to the results of [36, Corollary 4], the HSIC with function $\psi$ can be formulated as

$$\text{HSIC}(X, Y) = \int_{\mathbb{R}^{d_x} \times \mathbb{R}^{d_y}} \big|\phi_{\mathbb{P}_{XY}}(\omega) - \phi_{\mathbb{P}_X \mathbb{P}_Y}(\omega)\big|^2 (\mathcal{F}^{-1}\psi)(\omega)d\omega, \quad (4)$$

where $\omega = (\omega_x, \omega_y) \in \mathbb{R}^{d_x} \times \mathbb{R}^{d_y}$, $\omega_x, \omega_y$ are the frequencies of $X$ and $Y$ respectively, and

$$\phi_{\mathbb{P}_{XY}}(\omega) := \int e^{-i(\omega_x^T x + \omega_y^T y)} d\mathbb{P}_{XY}, \ \phi_{\mathbb{P}_X \mathbb{P}_Y}(\omega) := \left(\int e^{-i\omega_x^T x} d\mathbb{P}_X\right)\left(\int e^{-i\omega_y^T y} d\mathbb{P}_Y\right) \quad (5)$$

are the characteristic functions of $\mathbb{P}_{XY}$ and $\mathbb{P}_X \mathbb{P}_Y$, respectively. Intuitively, Eq. (4) means that HSIC can be understood as the difference between the joint distribution and the product of the marginal distributions in the frequency domain, with different weights $(\mathcal{F}^{-1}\psi)(\omega)$ being attached to different frequencies, which are determined by the kernel function. When $\mathcal{F}^{-1}\psi$ is almost everywhere non-zero, it can be shown that the kernel is characteristic [36, 10]. The characteristic condition ensures that the criterion is discriminative for discrepancies at almost all frequencies. However, with inappropriate choices of $\mathcal{F}^{-1}\psi$, the differences may not be significant enough. We explain this with an example:

**Example.** Consider the Sinusoid model that $\mathcal{X} \times \mathcal{Y} := [-\pi, \pi]^2$ and $(X, Y) \sim p_{xy}(x, y) \propto 1 + \sin(\omega_0 x)\sin(\omega_0 y)$, where $p_{xy}$ is the probability density function and $\omega_0$ is a positive integer. Combining Eq. (5), we can calculate that $\phi_{\mathbb{P}_X \mathbb{P}_Y}(\omega) = \delta(\omega_x)\delta(\omega_y)$ and $\phi_{\mathbb{P}_{XY}}(\omega) = \delta(\omega_x)\delta(\omega_y) + [\delta(\omega_x + \omega_0) + \delta(\omega_x - \omega_0)][\delta(\omega_y + \omega_0) + \delta(\omega_y - \omega_0)]$, where $\delta$ is the Dirac delta function, thus the difference between them only relies on the frequency $\omega_0$. When the Gaussian kernels with width $\sqrt{2}\lambda_x$ and $\sqrt{2}\lambda_y$ are used, i.e., $k(x, x') = \exp(-\|x - x'\|_2^2/(4\lambda_x^2))$, $l(y, y') = \exp(-\|y - y'\|_2^2/(4\lambda_y^2))$, then the inverse Fourier transform of $\psi$ is $(\mathcal{F}^{-1}\psi)(\omega_x, \omega_y) = \pi^{-1}\lambda_x\lambda_y \exp(-(\lambda_x^2\omega_x^2 + \lambda_y^2\omega_y^2))$. Hence HSIC$(X, Y) = 4\pi^{-1}\lambda_x\lambda_y \exp(-(\lambda_x^2 + \lambda_y^2)\omega_0^2)$ whose maximum is taken at $\lambda_x^* = \lambda_y^* = 1/(\sqrt{2}\omega_0)$, indicating that the widths need to be adjusted to focus on some specific frequencies. If the common setting [14, 44] is adapted, which uses mid-widths (i.e., the median distance does

not change with $\omega_0$ since the marginal distributions do not change with $\omega_0$), then the criterion will exponentially decline to 0 as $\omega_0$ increases. In contrast, the criterion using the adaptive optimization width $(1/\omega_0, 1/\omega_0)$ decreases at a rate of $\mathcal{O}(\omega_0^{-2})$, which is a considerable improvement.

This example illustrates the loss of the discriminatory power of the criterion when an inappropriate $\mathcal{F}^{-1}\psi$ is chosen. The discriminatory power of the criterion heavily impacts the sample size required for the test to obtain significant results in practice, and existing inflexible configurations may lead to inadequate test power in the presence of reasonably large sample sizes. Consequently, it is important to design learnable $\mathcal{F}^{-1}\psi$. To this end, we subsequently design a learnable objective and let it be optimized in a data-driven manner. Before this, we provide an approach to make the criterion be computed efficiently. This can be achieved by sampling in the frequency domain. Formally, a finite-dimensional approximation in the frequency domain of the integral in Eq. (4) is given as follows:

$$\text{HSIC}_\omega(X, Y) := \frac{1}{D_x D_y} \sum_{i=1}^{D_x} \sum_{j=1}^{D_y} \left| \phi_{\mathbb{P}_{XY}}(\omega_{x;i}, \omega_{y;j}) - \phi_{\mathbb{P}_X \mathbb{P}_Y}(\omega_{x;i}, \omega_{y;j}) \right|^2, \tag{6}$$

where $\{\omega_{x;i}\}_{i=1}^{D_x}, \{\omega_{y;j}\}_{j=1}^{D_y}$ are sampled independently with the measure $\mathcal{F}^{-1}\psi_k, \mathcal{F}^{-1}\psi_l$, respectively. Note that $\mathcal{F}^{-1}\psi$ is a product measure, i.e., $\mathcal{F}^{-1}\psi = (\mathcal{F}^{-1}\psi_k) \otimes (\mathcal{F}^{-1}\psi_l)$. This type of approximation is also called random Fourier features (RFF) [28] that had been applied to various kernel algorithms. We will incorporate this technique to efficiently perform computation later.

## 4 Learning Significant Fourier Feature Pairs

### 4.1 HSIC with Learnable Fourier Feature Pairs

To design $\mathcal{F}^{-1}\psi$, we need to make sure that $\text{supp}(\mathcal{F}^{-1}\psi) = \mathbb{R}^{d_x} \times \mathbb{R}^{d_y}$ to meet the characteristic condition and that its integral over the full space is 1 to ensure it is a probability measure. Also, for practical utility, $\mathcal{F}^{-1}\psi$ should embody a familiar probability density function, facilitating sampling procedures. Fortunately, a versatile array of options emerges through the judicious selection of kernels [2] with adjustable parameters. Take kernel $k$ as an example, some commonly used kernels are listed in Tab. 1, and their inverse Fourier transforms are listed simultaneously. Additionally, to

Table 1: Some popular kernels (parameterized by $\sigma, \Sigma$) with corresponding density functions.

| Kernel | $\psi_k(\Delta)$ | $\mathcal{F}^{-1}\psi_k(\omega)$ | $\mathcal{T}_{\theta_k}(x)$ | $p_k(\omega)$ |
|---|---|---|---|---|
| Gaussian | $e^{-\frac{\|\Delta\|_2^2}{2\sigma^2}}$ | $(2\pi)^{-d_x/2}\sigma e^{-\sigma^2\|\omega\|_2^2/2}$ | $x/\sigma$ | $(2\pi)^{-d_x/2}e^{-\|\omega\|_2^2/2}$ |
| Laplace | $e^{-\frac{\|\Delta\|_1}{\sigma}}$ | $\sqrt{\frac{2}{\pi}}\prod_d \frac{\sigma}{\sigma^2+\omega_d^2}$ | $x/\sigma$ | $\sqrt{\frac{2}{\pi}}\prod_d \frac{1}{1+\omega_d^2}$ |
| Mahalanobis | $e^{-\frac{1}{2}\Delta^T\Sigma^{-1}\Delta}$ | $(2\pi)^{-d_x/2}|\Sigma|^{-1/2}e^{-\omega^T\Sigma^{-1}\omega/2}$ | $\Sigma^{1/2}x$ | $(2\pi)^{-d_x/2}e^{-\|\omega\|_2^2/2}$ |

be able to apply gradient-based optimization techniques, we invoke a method that disentangle the sampled objects and the learnable parameters. Specifically, we leverage a variable transform $\mathcal{T}_{\theta_k}$ (a bijection function parameterized with $\theta_k$) to convert the probability measure $\mathcal{F}^{-1}\psi_k$ into a simple distribution (e.g. a standard Gaussian distribution) $p_k(\omega)$. Simultaneously, we relocate the learnable component onto $X$. Consequently, we can focus on learning parameterized transformations $\mathcal{T}_{\theta_k}$ and simplifying the computation by enabling sampling directly from $p_k(\omega)$.

**Remark.** The above scheme provides a broader form for designing. The mapping $\mathcal{T}_{\theta_k}$ can be viewed as a feature extractor, which makes it possible to flexibly combine models (e.g., neural network) thus incorporating deep kernel [24] into the framework. Also, it should be noted that the single kernel example can also be extended to multi-kernel setting [11] by executing the procedure for each kernel.

Next, we obtain the learnable independence criterion and utilize the sampling technique as in Eq. (6) to compute efficiently. Note that for simplicity, we take the same value for both $D_x$ and $D_y$ in Eq. (6) by default. By the definition, the kernel function can be expressed as

$$\psi_k\left(\mathcal{T}_{\theta_k}x - \mathcal{T}_{\theta_k}x'\right) = \mathcal{F}[\mathcal{F}^{-1}\psi_k(\omega)] = \int e^{-i\omega^T(\mathcal{T}_{\theta_k}x - \mathcal{T}_{\theta_k}x')}p_k(\omega)d\omega. \tag{7}$$

---

[2]The bounded, continuous, translation-invariant kernel satisfies the characteristic condition [12].

By applying the frequency sampling technique, we obtain the approximation as

$$\psi_k^{(\omega)}\left(\mathcal{T}_{\theta_k}x - \mathcal{T}_{\theta_k}x'\right) := \frac{2}{D}\sum_{j=1}^{D/2} e^{-i\omega_{k;j}^T(\mathcal{T}_{\theta_k}x - \mathcal{T}_{\theta_k}x')} = \frac{2}{D}\sum_{j=1}^{D/2}\cos\left(\omega_{k;j}^T(\mathcal{T}_{\theta_k}x - \mathcal{T}_{\theta_k}x')\right), \quad (8)$$

where $\{\omega_{k;j}\}_{j=1}^{D/2}$ are sampled independently with distribution $p_k(\omega)$ and the last equation is because the kernel function is real. To get a more computationally tractable form, we define

$$\Lambda_k(x) := \sqrt{\frac{2}{D}}\left[\cos\left(\omega_1^T\mathcal{T}_{\theta_k}x\right), \sin\left(\omega_1^T\mathcal{T}_{\theta_k}x\right), ..., \cos\left(\omega_{D/2}^T\mathcal{T}_{\theta_k}x\right), \sin\left(\omega_{D/2}^T\mathcal{T}_{\theta_k}x\right)\right], \quad (9)$$

called learnable RFF of $k$ then Eq. (8) becomes $\psi_k^{(\omega)}\left(\mathcal{T}_{\theta_k}x - \mathcal{T}_{\theta_k}x'\right) = \Lambda_k(x)\Lambda_k(x')^T$. The expression with a similar form is also given in [44], with the difference that we have added learnable parts. For $Y$, we define the corresponding symbols by substituting $k$ for $l$ and $x$ for $y$. Also, for convenience, we default to keeping $Y$ and $X$ the same number of samples $D$ from here on. Then the HSIC with learnable RFF pairs can be obtained by replacing $k, l$ in Eq. (1) to $\psi_k^{(\omega)}, \psi_l^{(\omega)}$. Also, the corresponding estimator with sample $Z$ can be obtained by replacing $\mathbf{K}, \mathbf{L}$ in Eq. (2) to the matrices $\mathbf{\Lambda}_X\mathbf{\Lambda}_X^T, \mathbf{\Lambda}_Y\mathbf{\Lambda}_Y^T$, where $\mathbf{\Lambda}_X := [\Lambda_k(x_1); ...; \Lambda_k(x_n)]_{n\times D}$ and so as define for $\mathbf{\Lambda}_Y$. As a result,

$$\text{HSIC}_\omega(Z) := \frac{1}{n^2}\text{Tr}(\mathbf{\Lambda}_X\mathbf{\Lambda}_X^T\mathbf{H}\mathbf{\Lambda}_Y\mathbf{\Lambda}_Y^T\mathbf{H}) = \frac{1}{n^2}\text{Tr}(\mathbf{\Lambda}_X^T\mathbf{H}\mathbf{\Lambda}_Y\mathbf{\Lambda}_Y^T\mathbf{H}\mathbf{\Lambda}_X) = \frac{1}{n^2}\|\mathbf{\Lambda}_{Xc}^T\mathbf{\Lambda}_{Yc}\|_F^2, \quad (10)$$

where $\mathbf{\Lambda}_{Xc} := \mathbf{H}\mathbf{\Lambda}_X, \mathbf{\Lambda}_{Yc} := \mathbf{H}\mathbf{\Lambda}_Y$. The time complexity is analyzed as follows. Since the computation of the mapping $\mathcal{T}_{\theta_k}x$ depends on the specific design, here we default to analyzing the kernel case shown in Tab. 1. In this case, computing $\mathbf{\Lambda}_X, \mathbf{\Lambda}_Y$ requires $\mathcal{O}\left(nD(d_x + d_y)\right)$ time. Then calculate $\mathbf{\Lambda}_{Xc}, \mathbf{\Lambda}_{Yc}$ cost $\mathcal{O}(nD)$. After that, calculate $\text{HSIC}_\omega(Z)$ cost $\mathcal{O}(nD^2)$. Hence, the overall time complexity is $\mathcal{O}\left(nD(d_x + d_y + D)\right)$, i.e. the running time is linear with $n$.

## 4.2 Linear-time Optimization Objective

Next, we model the behavior of $\text{HSIC}_\omega(Z)$ to obtain an optimization objective for maximizing the power of the test. By utilizing the property that $\text{HSIC}_\omega(Z)$ is a V-statistic, we can extend the results [14, Theorem 1, 2] for $\text{HSIC}_\omega(Z)$, as shown in the following proposition with the proof given in the Appendix. To simplify, we denote $(x_i, y_i)$ as $z_i$ to represent the $i$-th sample and denote $\psi_k^{(\omega)}\left(\mathcal{T}_{\theta_k}x_t - \mathcal{T}_{\theta_k}x_u\right)$ as $k_{tu}^{(\omega)}$ and $\psi_l^{(\omega)}\left(\mathcal{T}_{\theta_l}y_t - \mathcal{T}_{\theta_l}y_u\right)$ as $l_{tu}^{(\omega)}$.

**Proposition 1** (Asymptotics). *Let $h_{ijqr}^{(\omega)} := \frac{1}{4!}\sum_{(t,u,v,w)}^{(i,j,q,r)} k_{tu}^{(\omega)}l_{tu}^{(\omega)} + k_{tu}^{(\omega)}l_{vw}^{(\omega)} - 2k_{uv}^{(\omega)}l_{tv}^{(\omega)}$, where the sum represents all ordered quadruples $(t, u, v, w)$ drawn without replacement from $(i, j, q, r)$. Then, Under the null hypothesis $\mathcal{H}_0$, $HSIC_\omega(Z)$ coverages in distribution to*

$$nHSIC_\omega(Z) \xrightarrow{d} \sum_{l=1}^{\infty}\lambda_l\chi_{1l}^2, \quad \lambda_l g_l(z_j) = \int_{z_i, z_q, z_r} h_{ijqr}^{(\omega)}g_l(z_i)dF_{z_i, z_q, z_r}, \quad (11)$$

*where $\chi_{11}^2, \chi_{12}^2, ...$ are independent $\chi_1^2$ variates and $\lambda_l$ is the solution to the eigenvalue problem as in the right of Eq. (11). Also, under the alternative $\mathcal{H}_1$, $HSIC_\omega(Z)$ converges in distribution as*

$$n^{\frac{1}{2}}\left(HSIC_\omega(Z) - \mathbf{E}_Z HSIC_\omega(Z)\right) \xrightarrow{d} \mathcal{N}(0, \sigma_\omega^2), \quad \sigma_\omega^2 := 16\left[\mathbf{E}_i(\mathbf{E}_{j,q,r}h_{ijqr}^{(\omega)})^2 - \left(\mathbf{E}_Z h_{ijqr}^{(\omega)}\right)^2\right] \quad (12)$$

*with the simplified notation $\mathbf{E}_{j,q,r} := \mathbf{E}_{z_j, z_q, z_r}$ and $\mathbf{E}_Z := \mathbf{E}_{z_i, z_j, z_q, z_r}$.*

According to Proposition 1, the power of the test with $\text{HSIC}_\omega$ can be formulated by

$$\mathbb{P}_{\mathcal{H}_1}\left(n\text{HSIC}_\omega(Z) > r_\omega\right) \rightarrow \Phi\left(\frac{n\mathbf{E}_Z\text{HSIC}_\omega(Z) - r_\omega}{\sqrt{n}\sigma_\omega}\right), \quad (13)$$

where $\Phi$ is the standard normal CDF and $r_\omega$ is the threshold, i.e. $(1 - \alpha)$-quantile of distribution given in Eq. (11) that exactly controls Type I error rate to the nominal level $\alpha$. Hence, to maximize the power of the test, a natural criterion is $[n\mathbf{E}_Z\text{HSIC}_\omega(Z) - r_\omega]/(\sqrt{n}\sigma_\omega)$. Next, we provide its estimation which can be computed in linear time.

We first consider obtaining the estimator of the numerator part. For the term $\mathbf{E}_Z \mathrm{HSIC}_\omega(Z)$, we can estimate it with $\mathrm{HSIC}_\omega(Z)$ as in Eq. (10). The estimation of the threshold $r_\omega$ poses a challenge, primarily stemming from the lack of an explicit expression for the distribution of the infinite sum of chi-square variables. One avenue to address this challenge involves employing the permutation method [2, 38] to simulate the distribution under $\mathcal{H}_0$. However, this method necessitates a significant number of shuffles to accurately approximate the distribution. Furthermore, even with the implementation of parallel schemes, it incurs memory costs proportional to the number of permutations, rendering it impractical for resource-constrained scenarios. Here, we adopt a lightweight approach in practice, leveraging the gamma approximation as proposed by [14]. A gamma distribution is uniquely determined by its first and second-order moments. For these two moments, we present their corresponding linear-time estimators in Theorem 1. As a result, we can obtain the $(1 - \alpha)$-quantile of the gamma distribution, denoted as $\widehat{c_\alpha}$, with estimated parameters $\gamma := \mathcal{E}_0^2/\mathcal{V}_0, \beta := \mathcal{V}_0/\mathcal{E}_0$ in linear time. Formally, with the term $\mathcal{E}_0$ and $\mathcal{V}_0$ defined in Theorem 1, $\widehat{c_\alpha}$ is calculated by

$$\mathcal{H}_0 : n\mathrm{HSIC}_\omega(Z) \sim \frac{x^{\gamma-1}e^{-x/\beta}}{\beta^\gamma \Gamma(\gamma)}, \gamma = \frac{\mathcal{E}_0^2}{\mathcal{V}_0}, \ \beta = \frac{\mathcal{V}_0}{\mathcal{E}_0}, \ \int_0^{\frac{\widehat{c_\alpha}}{\beta}} \frac{x^{\gamma-1}e^{-x}}{\Gamma(\gamma)} dx = 1 - \alpha, \quad (14)$$

where $\Gamma(\cdot)$ is the gamma function. By combining the way to estimate the gradients of $\widehat{c_\alpha}$ [30], we enable it for gradient-based optimization with automatic differentiation framework. As a result, we obtain a linear-time differentiable estimator of the numerator part.

**Theorem 1** (Linear-Time Estimators). *Under $\mathcal{H}_0$, the estimators of mean and variance with bias of $\mathcal{O}(n^{-1})$ to $\mathbf{E}_Z[n\mathit{HSIC}_\omega(Z)]$ and $\mathit{Var}_Z[n\mathit{HSIC}_\omega(Z)]$, denote as $\mathcal{E}_0$ and $\mathcal{V}_0$, respectively, are given by*

$$\mathcal{E}_0 := \frac{[\mathbf{1}^T \mathbf{\Lambda}_{Xc}^{\cdot 2} \mathbf{1}][\mathbf{1}^T \mathbf{\Lambda}_{Yc}^{\cdot 2} \mathbf{1}]}{(n-1)^2}, \mathcal{V}_0 := \frac{2n(n-4)(n-5)}{(n-1)(n-2)(n-3)} \frac{[\mathbf{1}^T(\mathbf{\Lambda}_{Xc}^T \mathbf{\Lambda}_{Xc})^{\cdot 2}\mathbf{1}][\mathbf{1}^T(\mathbf{\Lambda}_{Yc}^T \mathbf{\Lambda}_{Yc})^{\cdot 2}\mathbf{1}]}{n^4},$$

$$(15)$$

*where $()^{\cdot 2}$ is the entry-wise matrix power. Both $\mathcal{E}_0$ and $\mathcal{V}_0$ can be calculated in $\mathcal{O}(nD^2)$ time.*

For the remain term $\sigma_\omega$, we estimate it with $\widehat{\sigma}_\omega$ that $\widehat{\sigma}_\omega^2 := 16\left[\frac{1}{n}\sum_i(\frac{1}{n^3}\sum_{j,q,r} h_{ijqr}^{(\omega)})^2 - \mathrm{HSIC}_\omega^2(Z)\right]$. To calculate $\sum_{j,q,r} h_{ijqr}^{(\omega)}$, the straightforward way is to compute each item $h_{ijqr}^{(\omega)}$, which requires total $\mathcal{O}(n^4)$ of computation. Here we provide a way to enable it to be calculated in linear time by obtaining a matrix expression. The main result is given by

$$\sum_{j,q,r} h_{ijqr}^{(\omega)} = \frac{1}{2}\left[n\mathbf{1}^T\mathbf{A}\mathbf{1} + n^2(\mathbf{A}\mathbf{1})_i + (\mathbf{1}^T\mathbf{C})\mathbf{B}_i + (\mathbf{1}^T\mathbf{B})\mathbf{C}_i - n\mathbf{E}_i - n\mathbf{F}_i - n\mathbf{D}_i - \mathbf{1}^T\mathbf{D}\right],$$

$$(16)$$

where the definition of variables $\mathbf{A}$ to $\mathbf{F}$ with the calculation cost are given in the Fig. 1 and the derivation of Eq. (16) is given in the Appendix. By checking the complexity of the remaining matrix operations in Eq. (16), all the elements with index $i$ can be calculated in $\mathcal{O}(nD^2)$. Combining the results obtained before that $\mathrm{HSIC}_\omega(Z)$ can also be calculated in $\mathcal{O}(nD^2)$, thus calculating the term $\widehat{\sigma}_\omega$ cost $\mathcal{O}(nD^2)$ time. As a result, we obtain the overall linear-time optimization objective $J := [\mathrm{HSIC}_\omega(Z) - \widehat{c_\alpha}/n]/\widehat{\sigma}_\omega$, which is a clear contrast to the existing quadratic-time schemes [30].

Figure 1: The diagram shows the definition of the quantities in Eq. (16), with styles representing the time complexity of the computational process in the current box. $\odot$: the element-wise product.

## 4.3 The Overall Learning Framework

After obtaining the differentiable optimization objective $J$, we can perform the training process end-to-end. In this process, the overfitting issues may happen especially with insufficient samples, which could influence both Type I and II errors. If we use the same sample for testing, the Type I errors may be uncontrollable [22] when the overfitting issues happen. To address this, we adopt the split scheme as in [24, 18] to allow our tests to maintain validity (controllable Type I errors). The split ratio is set to 0.5 to facilitate the balance between the two. Apart from controlling Type I errors, we still want to mitigate overfitting issues as much as possible in order to generalize the optimized

Fourier feature pairs on test data thus improving the power of our tests. To this end, we select smooth function classes to control the model complexity (e.g., as measured by the VC dimension), specifically in this paper, we consider the two classes in Tab. 1 and implement them for experiments later. One is the choice of Gaussian classes that optimize the global scale, and the other we consider Mahalanobis classes and set $\Sigma$ to be $\mathrm{diag}(\sigma_1, ..., \sigma_d)$ for optimization, which corresponds to optimizing the scale in each dimension (which allows to capture high-frequency signals such as the edge in the image). These smooth choices also bring the advantage of interpretability [30] and it is experimentally proven that this simple choice is already able to handle most of the cases in different settings.

**More discussion about split strategy.** Currently, there are two major classes of approaches for adaptive independence tests. One involves selecting kernels from a finite/countable set (discrete scenario) and the other involves performing kernel parameter searches in a continuous space (continuous scenario). For the former case, some methods [22, 32] control Type I errors by applying techniques from the selective inference literature without data splitting. However, these methods cannot be directly applied to a continuous scenario due to the uncountable set of kernels involved. To the best of our knowledge, both our scheme and existing methods [24, 18] rely on data splitting for the continuous case. Designing methods to control Type I errors in the continuous case without sample splitting remains a challenging and significant problem for future research.

**Algorithm.** Our algorithm is outlined in Alg. 1. As a pre-processing step, we split the data into the training data $Z^{tr}$ and the testing data $Z^{te}$ (Line 1). The test contains two phases: 1) We learn the Fourier feature pairs with Adam [21] optimizer using full batches on $Z^{tr}$ (Lines 2-7). 2) With the learned Fourier feature pairs, we calculate the test statistic and threshold (Lines 8-10) to determine the independence (Lines 11) on $Z^{te}$. The overall time complexity is $\mathcal{O}\big(TnD(d_x + d_y + D)\big)$ and the space cost is $\mathcal{O}\big(n(d_x + d_y + D)\big)$ for storing the data as well as the Fourier feature pairs.

---

**Algorithm 1** The learning and testing framework

---

**Input:** samples $Z$ of $X, Y$, significance level $\alpha$, the number of Fourier feature $D$.
**Output:** $X \perp\!\!\!\perp Y$ or $X \not\perp\!\!\!\perp Y$.

 1: Split the data as $Z = Z^{tr} \cup Z^{te}$. Sampling $\{\omega_j\}_{j=1}^{D/2} = \{(\omega_{k;j}, \omega_{l;j})\}_{j=1}^{D/2}$ with $p(\omega)$.
 2: $\triangleleft$ **Learning significant Fourier feature pairs on $Z^{tr}$.**
 3: Initialize parameters $\theta_k, \theta_l$, set learning rate $\epsilon$, and set iteration steps $T$.
 4: **for** $t = 1, 2, ..., T$ **do**
 5:     Obtain learnable Fourier feature pairs $\mathbf{\Lambda}_X, \mathbf{\Lambda}_Y$ with parameters $\theta_k, \theta_l$ and $\{\omega_j\}_{j=1}^{D/2}$.
 6:     Calculate criterion $J$ with $\mathbf{\Lambda}_X, \mathbf{\Lambda}_Y$ then optimize $J$ with $(\theta_k, \theta_l) \leftarrow (\theta_k, \theta_l) + \epsilon\nabla_{(\theta_k, \theta_l)}J$.
 7: **end for**
 8: After training, obtain optimized parameters $\theta_k^*, \theta_l^*$.
 9: $\triangleleft$ **Testing with learned Fourier feature pairs on $Z^{te}$.**
10: Calculate the statistic $n^{te}\mathrm{HSIC}_\omega(Z^{te})$, threshold $\widehat{c_\alpha}(Z^{te})$ with parameters $\theta_k^*, \theta_l^*$ and $\{\omega_j\}_{j=1}^{D/2}$.
11: Return $X \not\perp\!\!\!\perp Y$ if $\widehat{c_\alpha}(Z^{te}) \leq n^{te}\mathrm{HSIC}_\omega(Z^{te})$ holds, otherwise $X \perp\!\!\!\perp Y$.

---

## 5   Theoretical Results

We first give the uniform bound results over a ball in parameter space which guarantees the convergence of our optimizing objective thus ensuring its effectiveness in modeling test power.

**Theorem 2** (Uniform Bound). *Let $\theta_k, \theta_l$ parameterize $\mathcal{T}_{\theta_k}, \mathcal{T}_{\theta_l}$ in Banach spaces of dimension $d_k, d_l$. And $\mathcal{T}_{\theta_k}, \mathcal{T}_{\theta_l}$ are Lipschitz to the parameters $\theta_k, \theta_l$ with the non-negative constant $L_k, L_l$, respectively. Let $\Theta_c$ be a set of $(\theta_k, \theta_l)$ for which $\sigma_\omega \geq c > 0$ with a positive constant $c$ and $\|\theta_k\| \leq R_{\theta_k}, \|\theta_l\| \leq R_{\theta_l}$. Let $r$ denote the threshold, i.e., $(1 - \alpha)$-quantile for the distribution in Eq. (11) and $r^{(n)}$ be the threshold with sample size $n$. Let $\{(\omega_{k;j}, \omega_{l;j})\}_{j=1}^{D/2}$ be the samplings of frequency with the sampling number $D$. Also, we define $R_{\omega_k} := \sup_j \|\omega_{k;j}\|, R_{\omega_l} := \sup_j \|\omega_{l;j}\|, d_s := \max\{d_k, d_l\}$ and $\xi_\omega := \mathrm{HSIC}_\omega(Z)$. Then with probability at least $1 - \delta$, we have*

$$\sup_{(\theta_k, \theta_l) \in \Theta_c} \left| \frac{\xi_\omega - r_\omega^{(n)}/n}{\widehat{\sigma}_\omega} - \frac{\mathbf{E}_Z\xi_\omega - r_\omega/n}{\sigma_\omega} \right| \sim \mathcal{O}\left( \left[ \sqrt{\frac{1}{n}\log\frac{1}{\delta}} + d_s\frac{\log n}{n} + \frac{R_{\omega_k}L_k + R_{\omega_l}L_l}{\sqrt{n}} \right] \right).$$

Next, we show the consistency of the tests, i.e. the power of the test tends to 1 as the sample size increases. Let the U-statistic of $\text{HSIC}_\omega(Z^{te})$ be $\text{HSIC}_\omega^{(u)}(Z^{te})$, then we have the following results.

**Theorem 3** (Consistency). *Let $\theta_k^*, \theta_l^*$ be the parameters after learning, $Z^{te}$ be the test samples of size $m$, when $\mathbf{E}_Z HSIC_\omega^{(u)}(Z^{te}) > 0$, then the probability of the Type II error*

$$\mathbb{P}\big(\text{Type II error}\big) = \mathbb{P}_{\mathcal{H}_1}\big(mHSIC_\omega(Z^{te}) \le r_\omega^{(m)} | \theta_k^*, \theta_l^*\big) \sim \mathcal{O}(m^{-1/2}). \tag{17}$$

*Let the mapping functions with learned parameters $\theta_k^*, \theta_l^*$ be $\mathcal{T}_{\theta_k^*}, \mathcal{T}_{\theta_l^*}$, and the corresponding range space be compact subsets of $\mathbb{R}^{d_{\mathcal{T}_x}}, \mathbb{R}^{d_{\mathcal{T}_y}}$, respectively. Also, the diameters of two range spaces are denoted by $diam(\mathcal{T}_{\theta_k^*}), diam(\mathcal{T}_{\theta_l^*})$, respectively. Let $\{(\omega_{k;j}, \omega_{l;j})\}_{j=1}^{D/2}$ be the frequency samplings with their second moment denoted by $\sigma_{\omega_k}^2 := \mathbf{E}_{p_k(\omega)}[\omega_{k;j}^T \omega_{k;j}], \sigma_{\omega_l}^2 := \mathbf{E}_{p_l(\omega)}[\omega_{l;j}^T \omega_{l;j}]$. Additionally, we denote $\xi_u := HSIC(X, Y)$, then under $\mathcal{H}_1$, we have $\mathbf{E}_Z HSIC_\omega^{(u)}(Z^{te}) > 0$ with any constant probability when $D = \Omega\Big(\frac{d_{\mathcal{T}_x} + d_{\mathcal{T}_y}}{\xi_u^2} \log \frac{\sigma_{\omega_k} diam(\mathcal{T}_{\theta_k^*}) + \sigma_{\omega_l} diam(\mathcal{T}_{\theta_l^*})}{\xi_u}\Big)$.*

This result can be understood in two parts. The first one is about consistency, i.e., the Type II error rate tends to 0 at the rate of $m^{-1/2}$ when condition $\mathbf{E}_Z\text{HSIC}_\omega^{(u)}(Z^{te}) > 0$ holds. The second part provides the condition when $\mathbf{E}_Z\text{HSIC}_\omega^{(u)}(Z^{te}) > 0$ holds, which requires sufficiently many frequency samplings. The theorem shows that the large value of the criterion $\text{HSIC}(X, Y)$ helps to reduce the required $D$. According to the results discussed in Sec. 3, there is an improvement in the criterion by finding the more significant features and thus helps to reduce the required $D$. To summarize, the significant features further help to guarantee the consistency of the test under the efficient requirements (smaller $D$). All proofs as well as additional results are given in the Appendix.

## 6  Performance Evaluation

We compare the following tests: distance-based statistic **dCor** [39], the original HSIC **QHSIC** [14], the copula-based method **RDC** [26], the three variants of HSIC **NyHSIC** [44], **FHSIC** [44], **BH-SIC** [44] and **HSICAgg** [32], **NFSIC** [18] as introduced in Sec. 1. Among them, dCor and QHSIC are $\mathcal{O}(n^2)$ tests. RDC is calculated in $O(n \log n)$ time and the rest are $\mathcal{O}(n)$ tests. A detailed description of the comparing methods is given in the Appendix. For our methods, We provide two variants as mentioned in Sec. 4.3. We name the Gaussian class case as **LFHSIC-G**, and name the Mahalanobis class case (and set $\Sigma$ as a diagonal matrix) **LFHSIC-M**. Additionally, for the comparative methods [30] that are relevant to us, due to their high time overhead and therefore inability to handle some settings of evaluation, we separately provide a comparison with our method under certain feasible experimental settings, the results are given in the Appendix.

**Experimental setup.** The significance level $\alpha$ is set to $0.05$. We use Gaussian kernels for both $X$ and $Y$ in all kernel-based methods. And QHSIC, RDC, NyHSIC, FHSIC, BHSIC are all with the kernel width being set to the Euclidean distance median of the samples. The number of random features $D$ for FHSIC, LFHSIC-G/M, the number of induced variables for NyHSIC, the block size for BHSIC as well as the number of sub-diagonals $R$ for HSICAgg are all kept consistent as recommended in [44, 32] for fair evaluation. Parameter settings for the rest of the methods follow the defaults in the code. More details of the setups are given in the Appendix.

**Evaluation protocol.** We evaluate on four synthetic datasets [18, 30] and two real datasets [44, 30]. Synthetic datasets consist of Sine Dependency (SD), Sinusoid (Sin), Gaussian Sign (GSign), and independent subspace analysis (ISA) dataset [14]. On real data, we introduce high-dimensional image data and another music dataset to evaluate the capability of all methods in different data scenarios. Unless otherwise specified, we perform 100 repeated randomized experiments and report the average result of test power as default. More details of the generating process of each dataset and the details of the evaluation (including running time) are provided in the Appendix.

### 6.1  Results on Synthetic Datasets

**Settings of SD, Sin, and GSign Dataset**. The Sin data corresponds to the example in Sec. 3 that requires the method to focus on differences in specific frequencies. In SD, $Y$ is dependent solely on the first two dimensions of $X$. In contrast, in GSign, $Y$ is independent of any proper subset of $X$

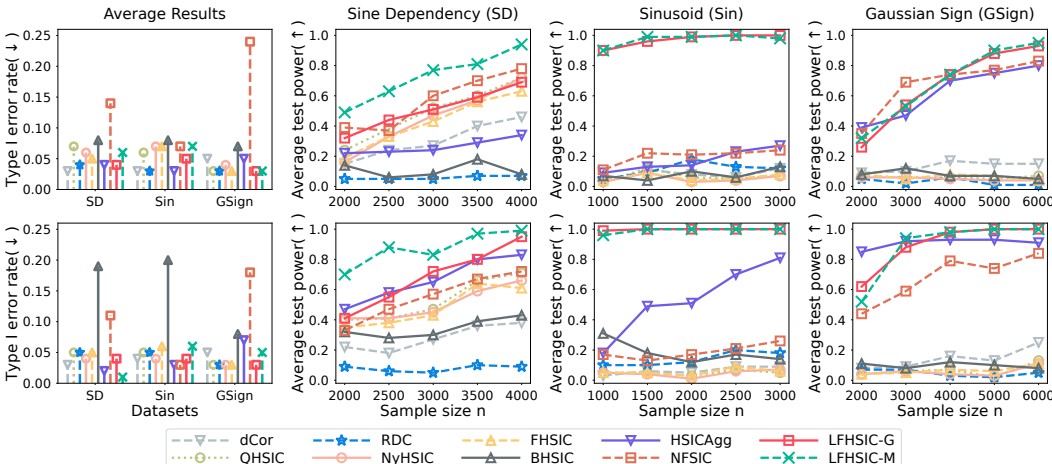

Figure 2: Top: ($D = 100$). Below: ($D = 500$). Left: The average Type I error rate on SD, Sin, and GSign datasets. The other three plots: The results of average test power on these three datasets.

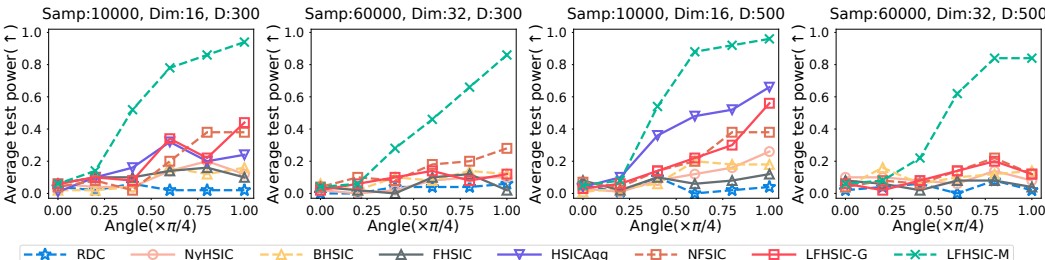

Figure 3: The average test power v.s. the rotation angle of each method on the ISA dataset.

but dependent on $X$ as a whole. Therefore, it requires the method to learn important local/global features based on the characteristics of the data to improve the test power. For SD and GSign, we set the dimension of $X$ as $4$ and $5$, respectively, and the dimension of $Y$ is $1$ for both. For Sin, we set the frequency parameter $\omega = 5$. For calculating the Type I error rate, we evaluate using samples ($n = 2000$) obtained by permutation for all three datasets.

**Performance.** The results for $D = 100$ and $D = 500$ are shown in Fig. 2. Except for NFSIC and BHSIC, all the other methods succeed in controlling the Type I error rate $\leq 0.05$. LFHSIC-M/G, NFHSIC, and HSICAgg perform much better than other methods due to their ability to obtain more appropriate kernels/features for testing. LFHSIC-G/M performs on both settings of $D$ and has a more significant advantage over the others when $D$ is small, implying the optimization objective can still be successfully optimized and the criterion is still powerful under high-speed requirements. In addition, as the sample size increases the test power of LFHSIC-G/M is gradually converging to $1$ in both settings, which corroborates the results of Theorem 3.

**Settings of ISA Dataset (Large Scale).** We set dimension (of both $X, Y$) and sample size as $d = 16, n = 10000$ and $d = 32, n = 60000$, then evaluate the average test power with angle parameter $\theta \in [0, \pi/4]$. Note that a larger angle signifies stronger dependency. The quadratic-time methods are not involved in the evaluation due to their inability to handle large-scale settings. For HSICAgg under the challenging setting $n = 60000, d = 32$, the memory space required for parallel implementation leads to memory overflow and hence the results are not given.

**Performance.** The results for $D = 300$ and $D = 500$ are shown in Fig. 3. The results obtained at $\theta = 0$ reflect the Type I error rate. All methods successfully control the Type I error rate $\leq 0.05$. LFHSIC-M stably outperforms other methods significantly as the angle increases. Method (LFHSIC-G) that simply optimizes the global bandwidth performs worse as $d$ increases, corroborating the need for more flexible kernel designs for more challenging tasks. Furthermore, comparing the

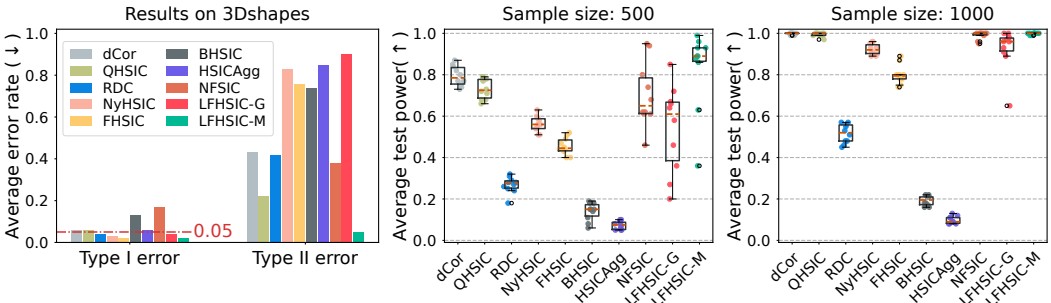

Figure 4: The results on two real data. Left: 3DShapes. Right two: MSD Dataset.

results under different settings of $D$, our method LFHSIC-M performs consistently well and exhibits progressively better performance as $D$ increases.

## 6.2 Results on Real Data

**Settings of Two Real Data**. The first real dataset used is a high-dimensional image dataset 3Dshapes as in [30]. In our experiments, we vectorize image $X$ to a vector with dimension $64 \times 64 \times 3 = 12,288$. The sample size is set as $128$. We add standard Gaussian noise $\mathcal{N}(0,1)$ to the angle $Y$ to make the setting more challenging. The Type I error rate is evaluated by the samples obtained by permutation. Besides, we consider the Million Song Data (MSD) as the second real dataset. The first dimension represents the year of release of each song and is referred to as variable $Y$. The remaining 90-dimensional features (e.g., mean timbre and timbre covariance) constitute variable $X$. We follow the recommended setting [44], i.e., disturbing each entry of the $X$ with an independent Gaussian noise $\mathcal{N}(0,1000)$. For this dataset MSD, in order to fully utilize the data, we randomly select $n \in \{500, 1000\}$ samples as the training set and other $n$ samples from the remaining data 100 times for the evaluation and obtain the average result. The above training and testing processes are repeated 10 times to evaluate the robustness of the optimization scheme.

**Performance.** The results of two real data with $D = 10$ are presented in Fig. 4. For the results on 3Dshapes (shown in the left of Fig. 4), all methods except BHSIC and NFSIC control the Type I error well. The linear-time test has relatively lower power compared to the quadratic-time test except for LFHSIC-M, proving that its more significant features obtained in high-dimensional scenarios enable it to achieve outstanding performance even in scenarios with high approximation requirements ($D = 10$). Similar conclusions can drawn from the MSD dataset (shown in the right of Fig. 4). Additionally, the results for NFSIC and LFHSIC-G/M with different sample sizes indicate increased robustness of the optimization as the sample size increases (reflected in the reduction of variance), and the more flexible design also contributes to this (comparing LFHSIC-M and LFHSIC-G), thus can be more effectively applied to real-world scenarios.

## 7 Conclusion

In this paper, we propose a novel method to efficiently learn significant Fourier feature pairs for maximizing the power of HSIC-based independence tests. By integrating a learnable Fourier feature module, we improve the flexibility of existing configurations and design a new criterion. The proposed linear-time optimization objective accurately models the power of the test and can be trained end-to-end in a data-driven manner, ensuring both effectiveness and efficiency. Both theoretical results and experimental results show the effectiveness of our proposed method. Future work includes further improving the sampling method in the frequency domain.

## Acknowledgments and Disclosure of Funding

This work was supported by National Natural Science Foundation (NSFC) (62372116), and National Key Research and Development Program of China (2021YFC3340302 and 2021YFC3300304).

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

# Appendix Organization

## A   List of Symbols and Notations

| | |
|---|---|
| $\mathcal{O}$ | big O notion |
| $o$ | small O notion |
| $i.i.d.$ | independent and identically distributed |
| $\mathbb{R}$ | the set of real numbers |
| $\mathcal{B}(\mathbb{R})$ | Borel $\sigma$-algebra on $\mathbb{R}$ |
| $\mathbb{P}_X$ | marginal distribution of $X$ |
| $\mathbb{P}_{XY}$ | joint distribution of $X, Y$ |
| $F_X$ | distribution function of $X$ |
| $\mathbf{E}[X]$ | expectation of $X$ |
| $\mathrm{Var}(X)$ | variance of $X$ |
| $X \perp\!\!\!\perp Y$ | random variables $X, Y$ are independent |
| $X \not\perp\!\!\!\perp Y$ | random variables $X, Y$ are not independent |
| $\mathbf{i}_r^n$ | the set of all $r$-tuples drawn without replacement from the set $\{1, ..., n\}$ |
| $\binom{n}{k}$ | number of $k$-combinations of $n$ elements |
| $(n)_k$ | number of permutations, define as $\frac{n!}{(n-k)!}$ |
| $\mathrm{Tr}(\cdot)$ | the trace of a square matrix |
| $\mathbf{1}$ | an vector of all ones |
| $\mathbf{H}$ | centering matrix define as $\mathbf{H} = \mathbf{I} - \frac{1}{n}\mathbf{1}\mathbf{1}^T$ |
| $\odot$ | element-wise product |
| $()^{\cdot 2}$ | element-wise power |
| $\xrightarrow{d}$ | convergence in distribution |
| $\otimes$ | the product symbol of measure |
| $\times$ | the product symbol of topological space |
| $\mathcal{N}(\Theta, r)$ | covering number with radii $r$ for space $\Theta$ |
| $\mathcal{F}, \mathcal{F}^{-1}$ | the Fourier transform, Fourier inverse transform |

## B   Assumptions

The following are the assumptions required. We denote the parameter spaces of $\theta_k, \theta_l$ as $\Theta_k, \Theta_l$.

(a) The mapping functions $\mathcal{T}_{\theta_k}, \mathcal{T}_{\theta_l}$ are Lipschitz to the parameters $\theta_k, \theta_l$, i.e. for all $x \in \mathcal{X}, y \in \mathcal{Y}$ and for all $\theta_k, \theta_k' \in \Theta_k, \theta_l, \theta_l' \in \Theta_l$,

$$\left\|\mathcal{T}_{\theta_k}(x) - \mathcal{T}_{\theta_k'}(x)\right\| \le L_k \cdot \|\theta_k - \theta_k'\|, \ \left\|\mathcal{T}_{\theta_l}(y) - \mathcal{T}_{\theta_l'}(y)\right\| \le L_l \cdot \|\theta_l - \theta_l'\| \tag{18}$$

with the nonnegative Lipschitz constant $L_k, L_l$.

(b) The range of the mapping functions $\mathcal{T}_{\theta_k}, \mathcal{T}_{\theta_l}$ are bounded.

(c) The parameters $\theta_0, \theta_1$ lie in Banach spaces of dimension $d_k, d_l$ respectively. Also, the parameters $\theta_k, \theta_l$ are bounded by $R_{\theta_k}, R_{\theta_l}$ respectively, i.e., $\|\theta_k\| \le R_{\theta_k}, \|\theta_l\| \le R_{\theta_l}$.

# C  Some Auxiliary Lemma

## C.1  A Useful Expression

In this part, we give a useful expression for $h_{ijqr}^{(\omega)}$ for the subsequent proof. By the definition, we have $h_{ijqr}^{(\omega)} = \frac{1}{4!} \sum_{(t,u,v,w)}^{(i,j,q,r)} k_{tu}^{(\omega)} l_{tu}^{(\omega)} + k_{tu}^{(\omega)} l_{vw}^{(\omega)} - 2k_{tu}^{(\omega)} l_{tv}^{(\omega)}$. We simplify it by setting $t = i$, $u = i$, $v = i$ and $w = i$ in turn. Then we can show that $h_{ijqr}^{(\omega)}$ is equal to

$$
\frac{1}{4!} \sum_{(u,v,w)}^{(j,q,r)} (k_{iu}^{(\omega)} l_{iu}^{(\omega)} + k_{iu}^{(\omega)} l_{vw}^{(\omega)} - 2k_{iu}^{(\omega)} l_{iv}^{(\omega)}) + \frac{1}{4!} \sum_{(t,v,w)}^{(j,q,r)} (k_{ti}^{(\omega)} l_{ti}^{(\omega)} + k_{ti}^{(\omega)} l_{vw}^{(\omega)} - 2k_{ti}^{(\omega)} l_{tv}^{(\omega)})
$$
$$
+ \frac{1}{4!} \sum_{(t,u,w)}^{(j,q,r)} (k_{tu}^{(\omega)} l_{tu}^{(\omega)} + k_{tu}^{(\omega)} l_{iw}^{(\omega)} - 2k_{tu}^{(\omega)} l_{ti}^{(\omega)}) + \frac{1}{4!} \sum_{(t,u,v)}^{(j,q,r)} (k_{tu}^{(\omega)} l_{tu}^{(\omega)} + k_{tu}^{(\omega)} l_{vi}^{(\omega)} - 2k_{tu}^{(\omega)} l_{tv}^{(\omega)}). \tag{19}
$$

By the definition, $k_{tu}^{(\omega)} := \psi_k^{(\omega)}(\mathcal{T}_{\theta_k} x_t - \mathcal{T}_{\theta_k} x_u) = \Lambda_X(x_t)\Lambda_X(x_u)^T$ is symmetric, i.e. $k_{tu}^{(\omega)} = k_{ut}^{(\omega)}$. And so as $l_{tu}^{(\omega)}$. Hence we can merge the identical items (marked with the same color). As a result,

$$
h_{ijqr}^{(\omega)} = \frac{1}{4!} \sum_{(u,v,w)}^{(j,q,r)} (2k_{iu}^{(\omega)} l_{iu}^{(\omega)} + 2k_{iu}^{(\omega)} l_{vw}^{(\omega)} - 2k_{iu}^{(\omega)} l_{iv}^{(\omega)}) - \frac{1}{4!} \sum_{(t,v,w)}^{(j,q,r)} (2k_{ti}^{(\omega)} l_{tv}^{(\omega)})
$$
$$
+ \frac{1}{4!} \sum_{(t,u,w)}^{(j,q,r)} (2k_{tu}^{(\omega)} l_{tu}^{(\omega)} + 2k_{tu}^{(\omega)} l_{iw}^{(\omega)} - 2k_{tu}^{(\omega)} l_{ti}^{(\omega)}) - \frac{1}{4!} \sum_{(t,u,v)}^{(j,q,r)} (2k_{tu}^{(\omega)} l_{tv}^{(\omega)}). \tag{20}
$$

We will use Eq. (20) many times in subsequent proofs.

## C.2  Properties of Learnable Random Fourier Feature

Under the assumption (a), RFFs $\psi_k^{(\omega)}, \psi_l^{(\omega)}$ are Lipschitz to the parameters $\theta_k, \theta_l$. Formally,

**Lemma 1.** *(Lipschitz Property of Fourier Feature). Let $\mathcal{T}_{\theta_k}, \mathcal{T}_{\theta_l}$ be the mapping functions of $X, Y$ that are Lipschitz to the parameters $\theta_k, \theta_l$ with the non-negative constant $L_k, L_l$, respectively. Let $\{(\omega_{k;j}, \omega_{l;j})\}_{j=1}^{D/2}$ be the samplings of frequency with the sampling number $D$. Also, we define $R_{\omega_k} := \sup_j \|\omega_{k;j}\|, R_{\omega_l} := \sup_j \|\omega_{l;j}\|$, then for the RFFs $\psi_k^{(\omega)}, \psi_l^{(\omega)}$ with mapping functions $\mathcal{T}_{\theta_k}, \mathcal{T}_{\theta_l}$ and frequency samplings $\{(\omega_{k;j}, \omega_{l;j})\}_{j=1}^{D/2}$, for all $(x, x') \in \mathcal{X} \times \mathcal{X}, (y, y') \in \mathcal{Y} \times \mathcal{Y}$ and for all $\theta_k, \theta_k' \in \Theta_k, \theta_l, \theta_l' \in \Theta_l$, we have*

$$
\|\psi_k^{(\omega)}(\Delta_{x,x'}) - \psi_k^{(\omega)}(\Delta'_{x,x'})\| \leq 2R_{\omega_k} L_k \cdot \|\theta_k - \theta_k'\|,
$$
$$
\|\psi_l^{(\omega)}(\Delta_{y,y'}) - \psi_l^{(\omega)}(\Delta'_{y,y'})\| \leq 2R_{\omega_l} L_l \cdot \|\theta_l - \theta_l'\|, \tag{21}
$$

*where $\Delta_{x,x'} := \mathcal{T}_{\theta_k} x - \mathcal{T}_{\theta_k} x'$, $\Delta'_{x,x'} := \mathcal{T}_{\theta_k'} x - \mathcal{T}_{\theta_k'} x'$ and $\Delta_{y,y'}, \Delta'_{y,y'}$ are defined by analogy.*

*Proof.* We prove the result for $\psi_k^{(\omega)}$ only since the proof for $\psi_l^{(\omega)}$ can be obtained in the same way. We start by recall the definition $\psi_k^{(\omega)}(\Delta_{x,x'}) := \frac{2}{D} \sum_{j=1}^{D/2} \cos(\omega_{k;j}^T \Delta_{x,x'})$. Then

$$
\|\psi_k^{(\omega)}(\Delta_{x,x'}) - \psi_k^{(\omega)}(\Delta'_{x,x'})\| = \left\| \frac{2}{D} \sum_{j=1}^{D/2} \cos(\omega_{k;j}^T \Delta_{x,x'}) - \frac{2}{D} \sum_{j=1}^{D/2} \cos(\omega_{k;j}^T \Delta'_{x,x'}) \right\|
$$
$$
\leq \frac{2}{D} \sum_{j=1}^{D/2} \| \cos(\omega_{k;j}^T \Delta_{x,x'}) - \cos(\omega_{k;j}^T \Delta'_{x,x'}) \| \tag{22}
$$

Since the cosine function is bounded by $1$, by the mean value theorem, for fixed $j$, we have

$$
\| \cos(\omega_{k;j}^T \Delta_{x,x'}) - \cos(\omega_{k;j}^T \Delta'_{x,x'}) \| \leq |\omega_{k;j}^T \Delta_{x,x'} - \omega_{k;j}^T \Delta'_{x,x'}|. \tag{23}
$$

Then according to the Cauchy–Schwarz inequality,

$$|\omega_{k;j}^T \Delta_{x,x'} - \omega_{k;j}^T \Delta'_{x,x'}| \le \|\omega_{k;j}\| \cdot \|\Delta_{x,x'} - \Delta'_{x,x'}\|. \tag{24}$$

By the definition of $\Delta_{x,x'}$ and the Lipschitz property of the mapping functions $\mathcal{T}_{\theta_k}, \mathcal{T}_{\theta_l}$, we have

$$\begin{aligned}
\|\Delta_{x,x'} - \Delta'_{x,x'}\| &= \|(\mathcal{T}_{\theta_k} x - \mathcal{T}_{\theta_k} x') - (\mathcal{T}_{\theta'_k} x - \mathcal{T}_{\theta'_k} x')\| \\
&\le \|\mathcal{T}_{\theta_k} x - \mathcal{T}_{\theta'_k} x\| + \|\mathcal{T}_{\theta_k} x' - \mathcal{T}_{\theta'_k} x'\| \le 2L_k \cdot \|\theta_k - \theta'_k\|.
\end{aligned} \tag{25}$$

Combining the above results, we complete the proof. $\qquad\square$

Under the assumption (b), we can obtain the uniform convergence property as follows.

**Lemma 2.** *(Uniform Convergence of Fourier Features). Let the mapping function of $X, Y$ with parameters $\theta_k, \theta_l$ be $\mathcal{T}_{\theta_k}, \mathcal{T}_{\theta_l}$, and the corresponding range space be a compact subset of $\mathbb{R}^{d_{\mathcal{T}_x}}, \mathbb{R}^{d_{\mathcal{T}_y}}$, respectively. Also, the diameter of two range spaces is denoted by $diam(\mathcal{T}_{\theta_k}), diam(\mathcal{T}_{\theta_l})$, respectively. Let $\{(\omega_{k;j}, \omega_{l;j})\}_{j=1}^{D/2}$ be the samplings of frequency with the sampling number $D$, then for the RFFs with mapping functions $\mathcal{T}_{\theta_k}, \mathcal{T}_{\theta_l}$ and frequency samplings $\{(\omega_{k;j}, \omega_{l;j})\}_{j=1}^{D/2}$, we have*

$$\begin{aligned}
\mathbb{P}\left[\sup_{x,x' \in \mathcal{X}} |\Lambda_k(x)^T \Lambda_k(x') - k(x,x')| \ge \epsilon\right] &\le 2^8 \left(\frac{\sigma_{\omega_k} diam(\mathcal{T}_{\theta_k})}{\epsilon}\right)^2 \exp\left(-\frac{D\epsilon^2}{4(d_{\mathcal{T}_x} + 2)}\right), \\
\mathbb{P}\left[\sup_{y,y' \in \mathcal{Y}} |\Lambda_l(y)^T \Lambda_l(y') - l(y,y')| \ge \epsilon\right] &\le 2^8 \left(\frac{\sigma_{\omega_l} diam(\mathcal{T}_{\theta_l})}{\epsilon}\right)^2 \exp\left(-\frac{D\epsilon^2}{4(d_{\mathcal{T}_y} + 2)}\right),
\end{aligned} \tag{26}$$

*where the second moment of frequency samplings $\sigma_{\omega_k}^2 := \mathbf{E}_{p_k(\omega)}[\omega_{k;j}^T \omega_{k;j}], \sigma_{\omega_l}^2 := \mathbf{E}_{p_l(\omega)}[\omega_{l;j}^T \omega_{l;j}].$*

*Proof.* Based on the derivation of RFFs in Sec. 4.1, We can view it as if the frequency sampling process is performed after the range space is obtained. Since the convergence bounds of the sampling process can be obtained directly through the results of [28, Claim 1], by replacing the input space in [28, Claim 1] to the range space here, then this part of the proof can be completed. $\qquad\square$

**Remark.** Combining the technique in [37], the constants in bounds can be further improved.

### C.3 Approximation Error Bound

Let $\text{HSIC}_\omega^{(u)}(Z)$, also denoted as $\xi_\omega^{(u)}$, be the U-statistic that corresponding to $\text{HSIC}_\omega(Z)$, i.e., $\text{HSIC}_\omega^{(u)}(Z) := \frac{1}{(n)_4} \sum_{(i,j,q,r) \in \mathbf{i}_4^n} h_{ijqr}^{(\omega)}$. The population value of $\text{HSIC}_\omega^{(u)}(Z)$ is given by $\mathbf{E}_Z \xi_\omega^{(u)}$ which can be viewed as the result obtained after a frequency sampling approximation on $\text{HSIC}(X, Y)$ is performed. The bound of approximation error is given by the following Lemma.

**Lemma 3.** *(Approximation Error Bound). For simplify, we denote $\Lambda_k(x)^T \Lambda_k(x'), \Lambda_l(y)^T \Lambda_l(y')$ as $k^{(\omega)}(x,x'), l^{(\omega)}(y,y')$, respectively. Then we have*

$$|\mathbf{E}_Z \xi_\omega^{(u)} - HSIC(X,Y)| \le 4 \cdot \sup_{x,x' \in \mathcal{X}, y,y' \in \mathcal{Y}} |k^{(\omega)}(x,x') l^{(\omega)}(y,y') - k(x,x') l(y,y')|. \tag{27}$$

*Proof.* We first represent $\mathbf{E}_Z \xi_\omega^{(u)}$ in the form corresponding to Eq. (1), i.e.,

$$\begin{aligned}
\mathbf{E}_Z \xi_\omega^{(u)} = \mathbf{E}_{XX'YY'}\left[k^{(\omega)}(X,X') l^{(\omega)}(Y,Y')\right] &+ \mathbf{E}_{XX'}\left[k^{(\omega)}(X,X')\right] \mathbf{E}_{YY'}\left[l^{(\omega)}(Y,Y')\right] \\
&- 2\mathbf{E}_{X'Y'}\left[\mathbf{E}_X k^{(\omega)}(X,X') \mathbf{E}_Y l^{(\omega)}(Y,Y')\right].
\end{aligned} \tag{28}$$

Taking one of the items as an example and comparing it to the corresponding item in Eq. (1),

$$\begin{aligned}
&\left|\mathbf{E}_{X'Y'}\left[\mathbf{E}_X k^{(\omega)}(X,X') \mathbf{E}_Y l^{(\omega)}(Y,Y')\right] - \mathbf{E}_{X'Y'}\left[\mathbf{E}_X k(X,X') \mathbf{E}_Y l(Y,Y')\right]\right| \\
&\le \int_{X',Y'} \int_X \int_Y |k^{(\omega)}(X,X') l^{(\omega)}(Y,Y') - k(X,X') l(Y,Y')| d\mathbb{P}_X d\mathbb{P}_Y d\mathbb{P}_{X'Y'} \\
&\le \sup_{x,x' \in \mathcal{X}, y,y' \in \mathcal{Y}} |k^{(\omega)}(x,x') l^{(\omega)}(y,y') - k(x,x') l(y,y')|.
\end{aligned} \tag{29}$$

The results can be obtained for the other terms in a similar way, which completes the proof. $\qquad\square$

# D  Proof of Proposition 1

In this section, we give a proof of the Proposition 1. We first restate the Proposition 1 here.

**Proposition 1** (Asymptotics). *Let* $h_{ijqr}^{(\omega)} := \frac{1}{4!} \sum_{(t,u,v,w)}^{(i,j,q,r)} k_{tu}^{(\omega)} l_{tu}^{(\omega)} + k_{tu}^{(\omega)} l_{vw}^{(\omega)} - 2k_{uv}^{(\omega)} l_{tv}^{(\omega)}$, *where the sum represents all ordered quadruples* $(t,u,v,w)$ *drawn without replacement from* $(i,j,q,r)$. *Then, Under the null hypothesis* $\mathcal{H}_0$, $HSIC_\omega(Z)$ *coverages in distribution to*

$$nHSIC_\omega(Z) \xrightarrow{d} \sum_{l=1}^{\infty} \lambda_l \chi_{1l}^2, \quad \lambda_l g_l(z_j) = \int_{z_i,z_q,z_r} h_{ijqr}^{(\omega)} g_l(z_i) dF_{z_i,z_q,z_r}, \tag{30}$$

*where* $\chi_{11}^2, \chi_{12}^2, ...$ *are independent* $\chi_1^2$ *variates and* $\lambda_l$ *is the solution to the eigenvalue problem as in the right of Eq. (30). Also, under the alternative* $\mathcal{H}_1$, $HSIC_\omega(Z)$ *converges in distribution as*

$$n^{\frac{1}{2}} \Big( HSIC_\omega(Z) - \mathbf{E}_Z HSIC_\omega(Z) \Big) \xrightarrow{d} \mathcal{N}(0, \sigma_\omega^2), \quad \sigma_\omega^2 = 16 \Big[ \mathbf{E}_i (\mathbf{E}_{j,q,r} h_{ijqr}^{(\omega)})^2 - \big( \mathbf{E}_Z h_{ijqr}^{(\omega)} \big)^2 \Big] \tag{31}$$

*with the simplified notation* $\mathbf{E}_{j,q,r} := \mathbf{E}_{z_j,z_q,z_r}$ *and* $\mathbf{E}_Z := \mathbf{E}_{z_i,z_j,z_q,z_r}$.

*Proof.* The proof is mainly based on [34, Chapter 5]. A proof for a similar result has been given in [14, Theorem 1,2]. The difference is that we consider the asymptotic distributions of $HSIC_\omega(Z)$ that used learnable RFF thus is a function of frequency samplings while they consider $HSIC_b(Z)$. Thus some steps need to be modified.

**Step 1**: we show that $HSIC_\omega(Z)$ is a V-statistic, this can be done since it can be expressed as $HSIC_\omega(Z) = \frac{1}{n^4} \sum_{i,j,q,r} h_{ijqr}^{(\omega)}$. To facilitate the conclusions in [34] for the U-statistic, we define the U-statistic $HSIC_\omega^{(u)}(Z) := \frac{1}{(n)_4} \sum_{(i,j,q,r) \in \mathbf{i}_4^n} h_{ijqr}^{(\omega)}$ that corresponds to $HSIC_\omega(Z)$.

**Step 2**: Then we prove the result under $\mathcal{H}_0$. To begin with, we first show that $\mathbf{E}_{j,q,r} h_{ijqr}^{(\omega)} = 0$ with $(i,j,q,r) \in \mathbf{i}_4^n$ under $\mathcal{H}_0$. According to Eq. (20), we can calculate $\mathbf{E}_{j,q,r} h_{ijqr}^{(\omega)}$ as

$$
\begin{aligned}
\mathbf{E}_{j,q,r} h_{ijqr}^{(\omega)} =& \frac{2}{4!} \sum_{(u,v,w)}^{(j,q,r)} \mathbf{E}_{u,v,w}(k_{iu}^{(\omega)} l_{iu}^{(\omega)} + k_{iu}^{(\omega)} l_{vw}^{(\omega)} - k_{iu}^{(\omega)} l_{iv}^{(\omega)}) - \frac{2}{4!} \sum_{(t,v,w)}^{(j,q,r)} \mathbf{E}_{t,v,w}(k_{ti}^{(\omega)} l_{tv}^{(\omega)}) \\
&+ \frac{2}{4!} \sum_{(t,u,w)}^{(j,q,r)} \mathbf{E}_{t,u,w}(k_{tu}^{(\omega)} l_{tu}^{(\omega)} + k_{tu}^{(\omega)} l_{iw}^{(\omega)} - k_{tu}^{(\omega)} l_{ti}^{(\omega)}) - \frac{2}{4!} \sum_{(t,u,v)}^{(j,q,r)} \mathbf{E}_{t,u,v}(k_{tu}^{(\omega)} l_{tv}^{(\omega)}) \\
=& \frac{1}{2} \mathbf{E}_{u,v,w}^{i \neq u \neq v \neq w}(k_{iu}^{(\omega)} l_{iu}^{(\omega)} + k_{iu}^{(\omega)} l_{vw}^{(\omega)} - k_{iu}^{(\omega)} l_{iv}^{(\omega)}) - \frac{1}{2} \mathbf{E}_{t,v,w}^{i \neq t \neq v \neq w}(k_{ti}^{(\omega)} l_{tv}^{(\omega)}) \\
&+ \frac{1}{2} \mathbf{E}_{t,u,w}^{i \neq t \neq u \neq w}(k_{tu}^{(\omega)} l_{tu}^{(\omega)} + k_{tu}^{(\omega)} l_{iw}^{(\omega)} - k_{tu}^{(\omega)} l_{ti}^{(\omega)}) - \frac{1}{2} \mathbf{E}_{t,u,v}^{i \neq t \neq u \neq v}(k_{tu}^{(\omega)} l_{tv}^{(\omega)}),
\end{aligned}
\tag{32}
$$

where we define simplified notions $\mathbf{E}_{u,v,w}^{i \neq u \neq v \neq w}$ whose superscript indicates the restriction. For readability, we will require additional notations: $\mathbf{E}_x k_i^{(\omega)} := \mathbf{E}_u^{i \neq u} k_{iu}^{(\omega)}$ and $\mathbf{E}_x k^{(\omega)} := \mathbf{E}_{t,u}^{t \neq u} k_{tu}^{(\omega)}$ (the notations for $Y$ is defined by analogy). Under $\mathcal{H}_0$, $X$ and $Y$ are independence. Hence

$$
\begin{aligned}
2\mathbf{E}_{j,q,r} h_{ijqr}^{(\omega)} =& \mathbf{E}_x k_i^{(\omega)} \mathbf{E}_y l_i^{(\omega)} + \mathbf{E}_x k_i^{(\omega)} \mathbf{E}_y l^{(\omega)} - \mathbf{E}_x k_i^{(\omega)} \mathbf{E}_y l_i^{(\omega)} - \mathbf{E}_x k_i^{(\omega)} \mathbf{E}_y l^{(\omega)} \\
&+ \mathbf{E}_x k^{(\omega)} \mathbf{E}_y l^{(\omega)} + \mathbf{E}_x k^{(\omega)} \mathbf{E}_y l_i^{(\omega)} - \mathbf{E}_x k^{(\omega)} \mathbf{E}_y l_i^{(\omega)} - \mathbf{E}_x k^{(\omega)} \mathbf{E}_y l^{(\omega)} = 0.
\end{aligned}
\tag{33}
$$

Then combining the results [34, Section 5.5.2], we can prove Eq. (30).

**Step 3**: Next we prove the asymptotic distribution under $\mathcal{H}_1$. We only need to show that $|HSIC_\omega(Z) - HSIC_\omega^{(u)}(Z)| \sim \mathcal{O}(1/n)$. By the definition of $k_{tu}^{(\omega)}, l_{tu}^{(\omega)}$, we can check that $|k_{tu}^{(\omega)}| \leq 1, |l_{tu}^{(\omega)}| \leq 1$ for all $t, u$, thus $|h_{ijqr}^{(\omega)}| \leq 4$ for all $i, j, q, r$. Hence we have

$$|HSIC_\omega(Z) - HSIC_\omega^{(u)}(Z)| \leq \frac{n^4 - (n)_4}{n^4} \cdot 4 + \Big( \frac{1}{(n)_4} - \frac{1}{n^4} \Big) \cdot (n)_4 \cdot 4 \sim \mathcal{O}(1/n). \tag{34}$$

Combining the results [34, Section 5.5.1], we can prove Eq. (31). □

# E  Proof of Theorem 1

In this section, we give a proof of the Theorem 1. We first restate the Theorem 1 here.

**Theorem 1** (Linear-Time Estimators). *Under $\mathcal{H}_0$, the estimators of mean and variance with bias of $\mathcal{O}(n^{-1})$ to $\mathbf{E}_Z[n\mathit{HSIC}_\omega(Z)]$ and $\mathbf{Var}_Z[n\mathit{HSIC}_\omega(Z)]$, denote as $\mathcal{E}_0$ and $\mathcal{V}_0$, respectively, are given by*

$$\mathcal{E}_0 := \frac{[\mathbf{1}^T \mathbf{\Lambda}_{Xc}^{\cdot 2} \mathbf{1}][\mathbf{1}^T \mathbf{\Lambda}_{Yc}^{\cdot 2} \mathbf{1}]}{(n-1)^2}, \mathcal{V}_0 := \frac{2n(n-4)(n-5)}{(n-1)(n-2)(n-3)} \frac{[\mathbf{1}^T (\mathbf{\Lambda}_{Xc}^T \mathbf{\Lambda}_{Xc})^{\cdot 2} \mathbf{1}][\mathbf{1}^T (\mathbf{\Lambda}_{Yc}^T \mathbf{\Lambda}_{Yc})^{\cdot 2} \mathbf{1}]}{n^4},$$
(35)

*where $()^{\cdot 2}$ is the entrywise matrix power. Both $\mathcal{E}_0$ and $\mathcal{V}_0$ can be calculated in $\mathcal{O}(nD^2)$ time.*

*Proof.* We first prove the part of the mean. Recall the definition of $\mathit{HSIC}_\omega(Z)$, we have $\mathit{HSIC}_\omega(Z) = \frac{1}{n^4} \sum_{i,j,q,r} h_{ijqr}^{(\omega)}$. Hence $\mathbf{E}_Z[\mathit{HSIC}_\omega(Z)] = \frac{1}{n^4} \sum_{i,j,q,r} \mathbf{E}_Z h_{ijqr}^{(\omega)}$. When $(i,j,q,r) \in \mathbf{i}_4^n$, then we can show that under $\mathcal{H}_0$, $\mathbf{E}_{i,j,q,r} h_{ijqr}^{(\omega)} = 0$ by performing the same analysis as in Eqs. (32) and (33). Then we consider the case where exactly two elements of $i, j, q, r$ are the same, for a total of $6n(n-1)(n-2)$ terms. By the symmetry of $h_{ijqr}^{(\omega)}$, the expectation of these terms all take the same value, and here we take $h_{iiqr}^{(\omega)}$ as an example. According to Eq. (20), we can represent $h_{iiqr}^{(\omega)}$ as

$$
\begin{aligned}
h_{iiqr}^{(\omega)} =& \frac{2}{4!} \sum_{(u,v,w)}^{(i,q,r)} (k_{iu}^{(\omega)} l_{iu}^{(\omega)} + k_{iu}^{(\omega)} l_{vw}^{(\omega)} - k_{iu}^{(\omega)} l_{iv}^{(\omega)}) - \frac{2}{4!} \sum_{(t,v,w)}^{(i,q,r)} (k_{ti}^{(\omega)} l_{tv}^{(\omega)}) \\
&+ \frac{2}{4!} \sum_{(t,u,w)}^{(i,q,r)} (k_{tu}^{(\omega)} l_{tu}^{(\omega)} + k_{tu}^{(\omega)} l_{iw}^{(\omega)} - k_{tu}^{(\omega)} l_{ti}^{(\omega)}) - \frac{2}{4!} \sum_{(t,u,v)}^{(i,q,r)} (k_{tu}^{(\omega)} l_{tv}^{(\omega)}).
\end{aligned}
$$
(36)

Under $\mathcal{H}_0$, $X$ and $Y$ are independence. Take the expectation on both sides, we have

$$
\begin{aligned}
12\mathbf{E}_{i,q,r} h_{iiqr}^{(\omega)} =& (2 + 4\mathbf{E}_x k^{(\omega)} \mathbf{E}_y l^{(\omega)}) + (2\mathbf{E}_y l^{(\omega)} + 4\mathbf{E}_x k^{(\omega)} \mathbf{E}_y l^{(\omega)}) \\
&- (2\mathbf{E}_x k^{(\omega)} + 2\mathbf{E}_y l^{(\omega)} + 2\mathbf{E}_x k^{(\omega)} \mathbf{E}_y l^{(\omega)}) - (2\mathbf{E}_y l^{(\omega)} + 4\mathbf{E}_x k^{(\omega)} \mathbf{E}_y l^{(\omega)}) \\
&+ (6\mathbf{E}_x k^{(\omega)} \mathbf{E}_y l^{(\omega)}) + (2\mathbf{E}_x k^{(\omega)} + 4\mathbf{E}_x k^{(\omega)} \mathbf{E}_y l^{(\omega)}) \\
&- (2\mathbf{E}_x k^{(\omega)} + 4\mathbf{E}_x k^{(\omega)} \mathbf{E}_y l^{(\omega)}) - (6\mathbf{E}_x k^{(\omega)} \mathbf{E}_y l^{(\omega)}) \\
=& 2(1 - \mathbf{E}_x k^{(\omega)} - \mathbf{E}_y l^{(\omega)} + \mathbf{E}_x k^{(\omega)} \mathbf{E}_y l^{(\omega)}),
\end{aligned}
$$
(37)

where we define additional notation $\mathbf{E}_x k^{(\omega)} := \mathbf{E}_{t,u}^{t \neq u} k_{tu}^{(\omega)}$ (the notation for $Y$ is defined by analogy) and use $k_{tt}^\omega = l_{tt}^\omega = 1$. Hence in this case, the sum of the contributions of all terms to $\mathbf{E}_Z[n\mathit{HSIC}_\omega(Z)]$ is $(1 - \mathbf{E}_x k^{(\omega)} - \mathbf{E}_y l^{(\omega)} + \mathbf{E}_x k^{(\omega)} \mathbf{E}_y l^{(\omega)}) + \mathcal{O}(1/n)$. For the remaining terms, i.e., the case where at least three of $i, j, q, r$ are equal, combined with the boundedness of $h_{ijqr}^{(\omega)}$, we can conclude that the sum of their contributions is $\mathcal{O}(1/n)$. As a result, we have shown that

$$\mathbf{E}_Z[n\mathit{HSIC}_\omega(Z)] = (1 - \mathbf{E}_x k^{(\omega)})(1 - \mathbf{E}_y l^{(\omega)}) + \mathcal{O}(n^{-1}).$$
(38)

The unbiased estimators of $\mathbf{E}_x k^{(\omega)}, \mathbf{E}_y l^{(\omega)}$ are given by $\mathbf{1}^T (\mathbf{\Lambda}_X \mathbf{\Lambda}_X^T - \mathbf{I}_n) \mathbf{1}, \mathbf{1}^T (\mathbf{\Lambda}_Y \mathbf{\Lambda}_Y^T - \mathbf{I}_n) \mathbf{1}$, respectively. Hence under $\mathcal{H}_0$, we obtain the estimator of mean with bias of $\mathcal{O}(n^{-1})$ as

$$\left[1 - \frac{\mathbf{1}^T (\mathbf{\Lambda}_X \mathbf{\Lambda}_X^T - \mathbf{I}_n) \mathbf{1}}{n(n-1)}\right] \left[1 - \frac{\mathbf{1}^T (\mathbf{\Lambda}_Y \mathbf{\Lambda}_Y^T - \mathbf{I}_n) \mathbf{1}}{n(n-1)}\right] = \frac{[\mathbf{1}^T \mathbf{\Lambda}_{Xc}^{\cdot 2} \mathbf{1}][\mathbf{1}^T \mathbf{\Lambda}_{Yc}^{\cdot 2} \mathbf{1}]}{(n-1)^2}.$$
(39)

Next, we prove the part of the variance. We start by calculating $\mathbf{Var}_Z[n\mathit{HSIC}_\omega^{(u)}(Z)]$, where the U-statistic $\mathit{HSIC}_\omega^{(u)}(Z) := \frac{1}{(n)_4} \sum_{(i,j,q,r) \in \mathbf{i}_4^n} h_{ijqr}^{(\omega)}$. According to the results [34, Section 5.2.1, Lemma A], we have

$$\mathbf{Var}[\mathit{HSIC}_\omega^{(u)}(Z)] = \binom{n}{4}^{-1} \sum_{c=1}^4 \binom{4}{c} \binom{n-4}{4-c} \zeta_c = \frac{4\binom{n-4}{3}}{\binom{n}{4}} \zeta_1 + \frac{6\binom{n-4}{2}}{\binom{n}{4}} \zeta_2 + \mathcal{O}(n^{-3}),$$
(40)

where $\zeta_1 := \mathbf{E}_i\big(\mathbf{E}_{j,q,r}h_{ijqr}^{(\omega)}\big)^2$ and $\zeta_2 := \mathbf{E}_{i,j}\big(\mathbf{E}_{q,r}h_{ijqr}^{(\omega)}\big)^2$. Under $\mathcal{H}_0$, when $(i,j,q,r) \in \mathbf{i}_4^n$, we can show that $\mathbf{E}_{j,q,r}h_{ijqr}^{(\omega)} = 0$ by performing the same analysis as in Eqs. (32) and (33), thus $\zeta_1 = 0$. For calculating $\zeta_2$, we mainly focus on the term $\mathbf{E}_{q,r}h_{ijqr}^{(\omega)}$. We use Eq. (20) again,

$$
12h_{ijqr}^{(\omega)} = \sum_{(u,v,w)}^{(j,q,r)}(k_{iu}^{(\omega)}l_{iu}^{(\omega)} + k_{iu}^{(\omega)}l_{vw}^{(\omega)} - k_{iu}^{(\omega)}l_{iv}^{(\omega)}) - \sum_{(t,v,w)}^{(j,q,r)}(k_{ti}^{(\omega)}l_{tv}^{(\omega)}) \\
+ \sum_{(t,u,w)}^{(j,q,r)}(k_{tu}^{(\omega)}l_{tu}^{(\omega)} + k_{tu}^{(\omega)}l_{iw}^{(\omega)} - k_{tu}^{(\omega)}l_{ti}^{(\omega)}) - \sum_{(t,u,v)}^{(j,q,r)}(k_{tu}^{(\omega)}l_{tv}^{(\omega)}).
\tag{41}
$$

Under $\mathcal{H}_0$, $X$ and $Y$ are independence. Take the expectation $\mathbf{E}_{q,r}$ on both sides, we have

$$
\begin{aligned}
12\mathbf{E}_{q,r}h_{ijqr}^{(\omega)} &= (2k_{ij}^{(\omega)}l_{ij}^{(\omega)} + 4\mathbf{E}_x k_i^{(\omega)}\mathbf{E}_y l_i^{(\omega)}) + (2k_{ij}^{(\omega)}\mathbf{E}_y l^{(\omega)} + 4\mathbf{E}_x k_i^{(\omega)}\mathbf{E}_y l_j^{(\omega)}) \\
&\quad - (2k_{ij}^{(\omega)}\mathbf{E}_y l_i^{(\omega)} + 2\mathbf{E}_x k_i^{(\omega)}l_{ij}^{(\omega)} + 2\mathbf{E}_x k_i^{(\omega)}\mathbf{E}_y l_i^{(\omega)}) \\
&\quad - (2k_{ij}^{(\omega)}\mathbf{E}_y l_j^{(\omega)} + 2\mathbf{E}_x k_i^{(\omega)}\mathbf{E}_y l_j^{(\omega)} + 2\mathbf{E}_x k_i^{(\omega)}\mathbf{E}_y l^{(\omega)}) \\
&\quad + (4\mathbf{E}_x k_j^{(\omega)}\mathbf{E}_y l_j^{(\omega)} + 2\mathbf{E}_x k^{(\omega)}\mathbf{E}_y l^{(\omega)}) + (4\mathbf{E}_x k_j^{(\omega)}\mathbf{E}_y l_i^{(\omega)} + 2\mathbf{E}_x k^{(\omega)}l_{ij}^{(\omega)}) \\
&\quad - (2\mathbf{E}_x k_j^{(\omega)}\mathbf{E}_y l_{ij}^{(\omega)} + 2\mathbf{E}_x k_j^{(\omega)}\mathbf{E}_y l_i^{(\omega)} + 2\mathbf{E}_x k^{(\omega)}\mathbf{E}_y l_i^{(\omega)}) \\
&\quad - (2\mathbf{E}_x k_j^{(\omega)}\mathbf{E}_y l_j^{(\omega)} + 2\mathbf{E}_x k_j^{(\omega)}\mathbf{E}_y l^{(\omega)} + 2\mathbf{E}_x k^{(\omega)}\mathbf{E}_y l_j^{(\omega)}) \\
&= 2(k_{ij}^{(\omega)} - \mathbf{E}_x k_i^{(\omega)} - \mathbf{E}_x k_j^{(\omega)} + \mathbf{E}_x k^{(\omega)})(l_{ij}^{(\omega)} - \mathbf{E}_y l_i^{(\omega)} - \mathbf{E}_y l_j^{(\omega)} + \mathbf{E}_y l^{(\omega)}),
\end{aligned}
\tag{42}
$$

where we define additional notation $\mathbf{E}_x k^{(\omega)} := \mathbf{E}_{t,u}^{t\neq u}k_{tu}^{(\omega)}, \mathbf{E}_x k_j^{(\omega)} := \mathbf{E}_u^{j\neq u}k_{ju}^{(\omega)}$ (the notations for $Y$ are defined by analogy). For simplify, we denote $k_{c;ij}^{(\omega)} := k_{ij}^{(\omega)} - \mathbf{E}_x k_i^{(\omega)} - \mathbf{E}_x k_j^{(\omega)} + \mathbf{E}_x k^{(\omega)}$ and $l_{c;ij}^{(\omega)} := l_{ij}^{(\omega)} - \mathbf{E}_y l_i^{(\omega)} - \mathbf{E}_y l_j^{(\omega)} + \mathbf{E}_y l^{(\omega)}$. Hence combining Eq. (40), we have

$$
\mathbf{Var}[n\mathrm{HSIC}_\omega^{(u)}(Z)] = \frac{2n(n-4)(n-5)}{(n-1)(n-2)(n-3)}\mathbf{E}_{i,j}(k_{c;ij}^{(\omega)})^2 \cdot \mathbf{E}_{i,j}(l_{c;ij}^{(\omega)})^2 + \mathcal{O}(n^{-1}).
\tag{43}
$$

Since under $\mathcal{H}_0$, the bias lead by the difference terms between $\mathrm{HSIC}_\omega^{(u)}(Z)$ and $\mathrm{HSIC}_\omega(Z)$ vanish faster than Eq. (43), hence the variance of $\mathrm{HSIC}_\omega(Z)$ is identical. In the following part, we consider the empirical estimate of the leading term in Eq. (43). We estimate $k_{c;ij}^{(\omega)}$ with $(\mathbf{\Lambda}_{Xc}\mathbf{\Lambda}_{Xc}^T)_{ij}$, then the estimation of $\mathbf{E}_{i,j}(k_{c;ij}^{(\omega)})^2$ is given by

$$
\frac{1}{n^2}\sum_{i,j}\big[(\Lambda_k(x_i) - \overline{\Lambda}_k)(\Lambda_k(x_j) - \overline{\Lambda}_k)^T\big]^2 = \frac{1}{n^2}\sum_{i,j}(\mathbf{\Lambda}_{Xc}\mathbf{\Lambda}_{Xc}^T)_{ij}^2 = \frac{\mathbf{1}^T(\mathbf{\Lambda}_{Xc}\mathbf{\Lambda}_{Xc}^T)^{\cdot2}\mathbf{1}}{n^2},
\tag{44}
$$

where we define notions $\overline{\Lambda}_k := \frac{1}{n}\sum_{u=1}^n \Lambda_k(x_u)$. Since computing the value of $\mathbf{1}^T(\mathbf{\Lambda}_{Xc}\mathbf{\Lambda}_{Xc}^T)^{\cdot2}\mathbf{1}$ requires $\mathcal{O}(n^2)$ time complexity, we transform it into a more computationally tractable form. We perform the following calculating as

$$
\mathbf{1}^T(\mathbf{\Lambda}_{Xc}\mathbf{\Lambda}_{Xc}^T)^{\cdot2}\mathbf{1} = \mathrm{Tr}(\mathbf{\Lambda}_{Xc}\mathbf{\Lambda}_{Xc}^T\mathbf{\Lambda}_{Xc}\mathbf{\Lambda}_{Xc}^T) = \mathrm{Tr}(\mathbf{\Lambda}_{Xc}^T\mathbf{\Lambda}_{Xc}\mathbf{\Lambda}_{Xc}^T\mathbf{\Lambda}_{Xc}) = \mathbf{1}^T(\mathbf{\Lambda}_{Xc}^T\mathbf{\Lambda}_{Xc})^{\cdot2}\mathbf{1}.
\tag{45}
$$

Recall the definition of $\mathbf{\Lambda}_{Xc} := [\mathbf{H}\mathbf{\Lambda}_X]_{n\times D}$ that can be calculated in $\mathcal{O}(nD)$ time, thus the term $[\mathbf{\Lambda}_{Xc}^T\mathbf{\Lambda}_{Xc}]_{D\times D}$ can be calculated in $\mathcal{O}(nD + nD^2)$ time. As a result, we obtain the estimator $[\mathbf{1}^T(\mathbf{\Lambda}_{Xc}^T\mathbf{\Lambda}_{Xc})^{\cdot2}\mathbf{1}][\mathbf{1}^T(\mathbf{\Lambda}_{Yc}^T\mathbf{\Lambda}_{Yc})^{\cdot2}\mathbf{1}]$ for the term $\mathbf{E}_{i,j}(k_{c;ij}^{(\omega)})^2 \cdot \mathbf{E}_{i,j}(l_{c;ij}^{(\omega)})^2$ that can be calculated in $\mathcal{O}(nD^2)$ time. The only thing left to do is to determine the bias of the estimator. For readable, we define $\widehat{k}_{ij}^{(\omega)} := k_{ij}^{(\omega)}, \widehat{k}_i^{(\omega)} := \frac{1}{n}\sum_u k_{iu}^{(\omega)}$ and $\widehat{k}^{(\omega)} := \frac{1}{n^2}\sum_{u,v} k_{uv}^{(\omega)}$, then by removing the terms with $i = j$, a estimate with difference $\mathcal{O}(n^{-1})$ to Eq. (44) is given by

$$
\frac{1}{n(n-1)}\sum_{i\neq j}\left[\widehat{k}_{ij}^{(\omega)} - \widehat{k}_i^{(\omega)} - \widehat{k}_j^{(\omega)} + \widehat{k}^{(\omega)}\right]^2.
\tag{46}
$$

By comparing the difference between the expectation of Eq. (46) and $\mathbf{E}_{i,j}(k_{c;ij}^{(\omega)})^2$, we can show that this error is bound by $O(1/n)$. We illustrate this by taking one of the cross terms as an example and the other terms by analogy, as shown in the following,

$$
\begin{aligned}
\mathbf{E}\Big[\frac{1}{n(n-1)}\sum_{i\neq j}\widehat{k}_i^{(\omega)}\widehat{k}^{(\omega)}\Big] &= \frac{1}{n^3(n-1)}\mathbf{E}\Big[\sum_i\sum_{q,r}\sum_u k_{iu}^{(\omega)}k_{qr}^{(\omega)}\Big] \\
&= \frac{1}{(n)_4}\mathbf{E}\Big[\sum_{(i,q,r,u)\in\mathbf{i}_4^n}k_{iu}^{(\omega)}k_{qr}^{(\omega)}\Big]+\mathcal{O}(n^{-1}) = \mathbf{E}_x k_i^{(\omega)}\mathbf{E}_x k^{(\omega)} + \mathcal{O}(n^{-1}).
\end{aligned}
\tag{47}
$$

Similarly, we can obtain the results for $\mathbf{E}_{i,j}(l_{c;ij}^{(\omega)})^2$. As a result, we have shown that $\mathcal{V}_0$ is a estimator of $\mathbf{Var}_Z[n\mathrm{HSIC}_\omega(Z)]$ with bias $\mathcal{O}(n^{-1})$ thus complete the whole proof. $\square$

## F  Calculation of Eq. (16)

Here, we give the computational details of Eq. (16). We mark colors to indicate correspondences.

According to Eq. (20), we can calculate $\sum_{j,q,r}h_{ijqr}^{(\omega)}$ as

$$
\begin{aligned}
\sum_{j,q,r}h_{ijqr}^{(\omega)} &= \frac{1}{2}\sum_{u,v,w}(k_{iu}^{(\omega)}l_{iu}^{(\omega)}+k_{iu}^{(\omega)}l_{vw}^{(\omega)}-k_{iu}^{(\omega)}l_{iv}^{(\omega)})-\frac{1}{2}\sum_{t,v,w}(k_{ti}^{(\omega)}l_{tv}^{(\omega)}) \\
&+ \frac{1}{2}\sum_{t,u,w}(k_{tu}^{(\omega)}l_{tu}^{(\omega)}+k_{tu}^{(\omega)}l_{iw}^{(\omega)}-k_{tu}^{(\omega)}l_{ti}^{(\omega)})-\frac{1}{2}\sum_{t,u,v}(k_{tu}^{(\omega)}l_{tv}^{(\omega)}).
\end{aligned}
\tag{48}
$$

We can further represent Eq. (48) in matrices form as

$$
\begin{aligned}
\sum_{j,q,r}h_{ijqr}^{(\omega)} = \frac{1}{2}\Big[ &n^2(\boldsymbol{\Lambda}_X\boldsymbol{\Lambda}_X^T\boldsymbol{\Lambda}_Y\boldsymbol{\Lambda}_Y^T)_{i,i}+(\boldsymbol{\Lambda}_X\boldsymbol{\Lambda}_X^T\mathbf{1})_i(\mathbf{1}^T\boldsymbol{\Lambda}_Y\boldsymbol{\Lambda}_Y^T\mathbf{1})-n[(\boldsymbol{\Lambda}_X\boldsymbol{\Lambda}_X^T\mathbf{1})\odot(\boldsymbol{\Lambda}_Y\boldsymbol{\Lambda}_Y^T\mathbf{1})]_i \\
&+ n\mathrm{Tr}(\boldsymbol{\Lambda}_X\boldsymbol{\Lambda}_X^T\boldsymbol{\Lambda}_Y\boldsymbol{\Lambda}_Y^T)+(\boldsymbol{\Lambda}_Y\boldsymbol{\Lambda}_Y^T\mathbf{1})_i(\mathbf{1}^T\boldsymbol{\Lambda}_X\boldsymbol{\Lambda}_X^T\mathbf{1})-n(\boldsymbol{\Lambda}_Y\boldsymbol{\Lambda}_Y^T\boldsymbol{\Lambda}_X\boldsymbol{\Lambda}_X^T\mathbf{1})_i \\
&- n(\boldsymbol{\Lambda}_X\boldsymbol{\Lambda}_X^T\boldsymbol{\Lambda}_Y\boldsymbol{\Lambda}_Y^T\mathbf{1})_i-(\mathbf{1}^T\boldsymbol{\Lambda}_X\boldsymbol{\Lambda}_X^T\boldsymbol{\Lambda}_Y\boldsymbol{\Lambda}_Y^T\mathbf{1})\Big].
\end{aligned}
\tag{49}
$$

Next, by variable substitution, we obtain the result as

$$
\sum_{j,q,r}h_{ijqr}^{(\omega)} = \frac{1}{2}\left[n\mathbf{1}^T\mathbf{A}\mathbf{1}+n^2(\mathbf{A}\mathbf{1})_i+(\mathbf{1}^T\mathbf{C})\mathbf{B}_i+(\mathbf{1}^T\mathbf{B})\mathbf{C}_i-n\mathbf{E}_i-n\mathbf{F}_i-n\mathbf{D}_i-\mathbf{1}^T\mathbf{D}\right].
\tag{50}
$$

where the definition of variables $\mathbf{A}$ to $\mathbf{F}$ with the calculation cost are given in the Fig. 1. For convenience, we re-show the diagram here for reference.

Figure 5: The diagram shows the definition of the quantities, with styles representing the time complexity of the computational process in the current box. $\odot$: the element-wise product.

The computational complexity of each step is illustrated in Fig. 5. We explain some steps here. As a start, recall that the size of $\boldsymbol{\Lambda}_X, \boldsymbol{\Lambda}_Y$ are both $n\times D$. Therefore a time complexity $\mathcal{O}(nD^2)$ is required to compute $\boldsymbol{\Lambda}_X^T\boldsymbol{\Lambda}_Y$ by matrix multiplication operation. Further multiplying the obtained $[\boldsymbol{\Lambda}_X^T\boldsymbol{\Lambda}_Y]_{D\times D}$ with $\boldsymbol{\Lambda}_Y^T$ requires $\mathcal{O}(nD^2)$ time complexity. Next, since both $\boldsymbol{\Lambda}_X$ and $(\boldsymbol{\Lambda}_X^T\boldsymbol{\Lambda}_Y\boldsymbol{\Lambda}_Y^T)^T$ are of size $n\times D$, the elemental product operation requires a time complexity of $\mathcal{O}(nD)$. In a similar way, we can check the time complexity for each remained step. After getting the variables $\mathbf{A}$ to $\mathbf{F}$, since $\mathbf{1}^T\mathbf{A}\mathbf{1}, \mathbf{A}\mathbf{1}$ all can be calculated in $\mathcal{O}(nD)$ and $\mathbf{1}^T\mathbf{B}, \mathbf{1}^T\mathbf{C}, \mathbf{1}^T\mathbf{D}$ all can be calculated in $\mathcal{O}(n)$, we conclude that the results with index $i$ in Eq. (50) can be obtained in $\mathcal{O}(nD^2)$ time.

## G    Proof of Theorem 2

In this section, we give a proof of the Theorem 2. We first restate the Theorem 2 here.

**Theorem 2** (Uniform Bound). *Let $\theta_k, \theta_l$ parameterize $\mathcal{T}_{\theta_k}, \mathcal{T}_{\theta_l}$ in Banach spaces of dimension $d_k, d_l$. And $\mathcal{T}_{\theta_k}, \mathcal{T}_{\theta_l}$ are Lipschitz to the parameters $\theta_k, \theta_l$ with the non-negative constant $L_k, L_l$, respectively. Let $\Theta_c$ be a set of $(\theta_k, \theta_l)$ for which $\sigma_\omega \geq c > 0$ with a positive constant $c$ and $\|\theta_k\| \leq R_{\theta_k}, \|\theta_l\| \leq R_{\theta_l}$. Let $r$ denote the threshold, i.e., $(1 - \alpha)$-quantile for the distribution in Eq. (11) and $r^{(n)}$ be the threshold with sample size $n$. Let $\{(\omega_{k;j}, \omega_{l;j})\}_{j=1}^{D/2}$ be the samplings of frequency with the sampling number $D$. Also, we define $R_{\omega_k} := \sup_j \|\omega_{k;j}\|, R_{\omega_l} := \sup_j \|\omega_{l;j}\|, d_s := \max\{d_k, d_l\}$ and $\xi_\omega := HSIC_\omega(Z)$. Then with probability at least $1 - \delta$, we have*

$$\sup_{(\theta_k, \theta_l) \in \Theta_c} \left| \frac{\xi_\omega - r_\omega^{(n)}/n}{\widehat{\sigma}_\omega} - \frac{\mathbf{E}_Z \xi_\omega - r_\omega/n}{\sigma_\omega} \right| \sim \mathcal{O}\left( \left[ \sqrt{\frac{1}{n} \log \frac{1}{\delta}} + d_s \frac{\log n}{n} + \frac{R_{\omega_k} L_k + R_{\omega_l} L_l}{\sqrt{n}} \right] \right).$$

*Proof.* We take a similar roadmap of proof as [30] and extend it to our optimization objective. The roadmap of the proof is as follows: we first obtain the convergence results (with sample size $n$) for each estimator with fixed parameters $\theta_k, \theta_l$, and then extend the results to the entire parameter space via $\epsilon$-net arguments. We begin the proof of the first part, which is based on bounded differences inequality (McDiarmid's inequality) [41, Theorem 2.9.1].

**Bound of $|\xi_\omega - \mathbf{E}_Z \xi_\omega|$.** Recall the definition of $\xi_\omega := \text{HSIC}_\omega(Z) = \frac{1}{n^4} \sum_{i,j,q,r} h_{ijqr}^{(\omega)}$. By the definition of $k_{tu}^{(\omega)}, l_{tu}^{(\omega)}$, we have $|k_{tu}^{(\omega)}| \leq 1, |l_{tu}^{(\omega)}| \leq 1$ for all $t, u$, thus $|h_{ijqr}^{(\omega)}| \leq 4$ for all $i, j, q, r$. Now we begin by showing the bounded differences property of $h_{ijqr}^{(\omega)}$. Concretely, we replace the first sample $z_1 = (x_1, y_1)$ with $z_1' = (x_1', y_1')$ and keep the remaining samples the same. The obtained samples are named as $Z'$. Then the difference terms between $h_{ijqr}^{(\omega)}$ and the new substitution $\breve{h}_{ijqr}^{(\omega)}$ can only happen in the case that at least one of $i, j, q, r$ is equal to $1$. For the case that only one subscript is $1$ (here take $i = 1$ for example), combining Eq. (20), we have

$$\left| \sum_{j,q,r} h_{1jqr}^{(\omega)} - \sum_{j,q,r} \breve{h}_{1jqr}^{(\omega)} \right| \leq \frac{2}{4!} (n-1)(n-2)(n-3) \cdot 6 \cdot 16 = 8(n-1)(n-2)(n-3). \quad (51)$$

The whole contributes of remaining terms that at least two $i, j, q, r$ are less than $\mathcal{O}(n^{-2})$, thus

$$\left| \frac{1}{n^4} \sum_{i,j,q,r} h_{ijqr}^{(\omega)} - \frac{1}{n^4} \sum_{i,j,q,r} \breve{h}_{ijqr}^{(\omega)} \right| \leq 4 \cdot \left| \frac{1}{n^4} \sum_{j,q,r} h_{1jqr}^{(\omega)} - \frac{1}{n^4} \sum_{j,q,r} \breve{h}_{1jqr}^{(\omega)} \right| + \mathcal{O}(n^{-2}) = \mathcal{O}(n^{-1}). \quad (52)$$

Hence $\text{HSIC}_\omega(Z)$ satisfy the bounded differences property with $\mathcal{O}(n^{-1})$. Using McDiarmid's inequality, for fixed $\theta_0, \theta_1$, with probability at least $1 - \delta$, there exist a universal constant $C_1$ such that

$$\left| \xi_\omega - \mathbf{E}_Z \xi_\omega \right| \leq C_1 \sqrt{\frac{1}{n} \log \frac{2}{\delta}}. \quad (53)$$

**Bound of $|r_\omega^{(n)} - r_\omega|$.** As $r_\omega^{(n)}$ is the $(1 - \alpha)$ of the distribution of $n\xi_\omega$ with sample size $n$ under $\mathcal{H}_0$, according to the Eq. (53), when $n$ is large enough, there exist a universal constant $C_2$ such that

$$|r_\omega^{(n)}|/n \leq C_1 \sqrt{\frac{1}{n} \log \frac{2}{\alpha}} + |\mathbf{E}_Z \xi_\omega| \leq C_2 \sqrt{\frac{1}{n} \log \frac{1}{\alpha}}, \quad (54)$$

where the last inequation is because $\mathbf{E}_Z \xi_\omega \sim \mathcal{O}(n^{-1})$ under $\mathcal{H}_0$ (see Theorem 1 for a detailed explanation). Hence $|r_\omega^{(n)}| \sim \mathcal{O}\left( \sqrt{n \log(1/\alpha)} \right)$. Also, by definition $r_\omega$ is a constant related to $\alpha$.

**Bound of $|\widehat{\sigma}_\omega^2 - \sigma_\omega^2|$.** In this part, We first obtain the bound of $|\widehat{\sigma}_\omega^2 - \mathbf{E}_Z \widehat{\sigma}_\omega^2|$, then obtain the bound of $|\mathbf{E}_Z \widehat{\sigma}_\omega^2 - \sigma_\omega^2|$. As before we start by showing the bounded variance property of $\widehat{\sigma}_\omega^2$. We replace $z_1 = (x_1, y_1)$ with $z_1' = (x_1', y_1')$ and keep the remaining samples the same. The obtained samples are named as $Z'$. For readable, we denote $\widehat{\sigma}_\omega^2$ with sample $Z, Z'$ as $\widehat{\sigma}_\omega^2(Z), \widehat{\sigma}_\omega^2(Z')$ respectively.

Recall the definition $\widehat{\sigma}_\omega^2 := 16 \left[ \frac{1}{n} \sum_i \left( \frac{1}{n^3} \sum_{j,q,r} h_{ijqr}^{(\omega)} \right)^2 - \text{HSIC}_\omega^2(Z) \right]$. Since $|h_{ijqr}^{(\omega)}| \leq 4$, we have

$$\left| \frac{1}{n} \sum_i \left( \frac{1}{n^3} \sum_{j,q,r} h_{ijqr}^{(\omega)} \right)^2 - \frac{1}{n} \sum_i \left( \frac{1}{n^3} \sum_{j,q,r} \breve{h}_{ijqr}^{(\omega)} \right)^2 \right| \leq \frac{8}{n^4} \sum_i \sum_{j,q,r} |h_{ijqr}^{(\omega)} - \breve{h}_{ijqr}^{(\omega)}|, \quad (55)$$

$$\left|\left(\frac{1}{n^4}\sum_{i,j,q,r}h_{ijqr}^{(\omega)}\right)^2-\left(\frac{1}{n^4}\sum_{i,j,q,r}\breve{h}_{ijqr}^{(\omega)}\right)^2\right|\le\frac{8}{n^4}\sum_{i,j,q,r}\left|h_{ijqr}^{(\omega)}-\breve{h}_{ijqr}^{(\omega)}\right|. \tag{56}$$

Again, the difference terms between $h_{ijqr}^{(\omega)}$ and the new substitution $\breve{h}_{ijqr}^{(\omega)}$ can only happen in the case that at least one of $i,j,q,r$ is equal to 1. Hence, Eqs. (55) and (56) are both $\mathcal{O}(n^{-1})$. As a result, $\widehat{\sigma}_\omega^2$ satisfy the bounded differences property with bound $\mathcal{O}(n^{-1})$. Using McDiarmid's inequality, with probability at least $1-\delta$, there exist a universal constant $C_3$ such that $\left|\widehat{\sigma}_\omega^2-\mathbf{E}_Z\widehat{\sigma}_\omega^2\right|\le C_3\sqrt{\frac{1}{n}\log\frac{2}{\delta}}$. Next we obtain the bound of $|\mathbf{E}_Z\widehat{\sigma}_\omega^2-\sigma_\omega^2|$. We rewrite $\mathbf{E}_Z\widehat{\sigma}_\omega^2$ as

$$\mathbf{E}_Z\widehat{\sigma}_\omega^2=16\left(\frac{1}{n^7}\sum_{ijqrj'q'r'}\mathbf{E}[h_{ijqr}^{(\omega)}h_{ij'q'r'}^{(\omega)}]-\frac{1}{n^8}\sum_{ijqri'j'q'r'}\mathbf{E}[h_{ijqr}^{(\omega)}h_{i'j'q'r'}^{(\omega)}]\right). \tag{57}$$

By adding further restrictions that $i,i',j,q,r,j',q',r'$ are all different, we can obtain the corresponding expression for $\sigma_\omega^2$. Hence the difference between them can only happen when at least one subscript in $i',j,q,r,j',q',r'$ is equal to $i$. Combining $|h_{ijqr}^{(\omega)}|\le 4$, we have $|\mathbf{E}_Z\widehat{\sigma}_\omega^2-\sigma_\omega^2|\sim\mathcal{O}(n^{-1})$.

**$\epsilon$-net arguments.** Next, we prove the second part with $\epsilon$-net arguments. Take the parameter space $\Theta_k$ of $\theta_k$ as an example. We choose a cover with $\mathcal{N}(\Theta_k,r_k)$ points $\{p_i\}_{i=1}^{\mathcal{N}(\Theta_k,r_k)}$ such that for any point $p\in\Theta_k$, we have $\min_i\|p-p_i\|\le r_k$. According to [41, Proposition 4.2.12], by comparing the volumes, we have $\mathcal{N}(\Theta_k,r_k)\le(4R_{\Theta_k}/r_k)^{d_k}$. As for the parameter space $\Theta_l$ of $\theta_l$, we can also obtain a cover with $\mathcal{N}(\Theta_l,r_l)$ points that $\mathcal{N}(\Theta_k,r_k)\le(4R_{\Theta_l}/r_l)^{d_l}$. Here, we set $r_k=4R_{\Theta_k}/\sqrt{n},r_l=4R_{\Theta_l}/\sqrt{n}$, thus $\mathcal{N}(\Theta_k,r_k)\le(\sqrt{n})^{d_k},\mathcal{N}(\Theta_l,r_l)\le(\sqrt{n})^{d_l}$. Then combining the Lipschitz property as shown in Lemma 4, we have with probability at least $1-\delta$,

$$\sup_{(\theta_k,\theta_l)\in\Theta_c}\left|\xi_\omega-\mathbf{E}_Z\xi_\omega\right|\le C_1\sqrt{\frac{1}{n}\log\frac{2\mathcal{N}(\Theta_k,r_k)\mathcal{N}(\Theta_l,r_l)}{\delta}}+8R_{\omega_k}L_k\cdot r_k+8R_{\omega_l}L_lr_l$$

$$\le C_1\sqrt{\frac{1}{n}\log\frac{2}{\delta}+(d_k+d_l)\frac{\log n}{2n}}+\frac{32R_{\omega_k}L_kR_{\Theta_k}}{\sqrt{n}}+\frac{32R_{\omega_l}L_lR_{\Theta_l}}{\sqrt{n}}.$$

Hence when $n$ is large enough, there exists a positive constant $C_3$, with probability at least $1-\delta$,

$$\sup_{(\theta_k,\theta_l)\in\Theta_c}\left|\xi_\omega-\mathbf{E}_Z\xi_\omega\right|\le C_3\left[\sqrt{\frac{1}{n}\log\frac{1}{\delta}+(d_k+d_l)\frac{\log n}{n}}+\frac{R_{\omega_k}L_kR_{\Theta_k}}{\sqrt{n}}+\frac{R_{\omega_l}L_lR_{\Theta_l}}{\sqrt{n}}\right]. \tag{58}$$

Similar, when $n$ is large enough, there exists a positive constant $C_4$, with probability at least $1-\delta$,

$$\sup_{(\theta_k,\theta_l)\in\Theta_c}\left|\widehat{\sigma}_\omega^2-\sigma_\omega^2\right|\le C_4\left[\sqrt{\frac{1}{n}\log\frac{1}{\delta}+(d_k+d_l)\frac{\log n}{n}}+\frac{R_{\omega_k}L_kR_{\Theta_k}}{\sqrt{n}}+\frac{R_{\omega_l}L_lR_{\Theta_l}}{\sqrt{n}}\right]. \tag{59}$$

**Overall Bound.** Now we combine the previously obtained results. We have

$$\left|\frac{\xi_\omega-r_\omega^{(n)}/n}{\widehat{\sigma}_\omega}-\frac{\mathbf{E}_Z\xi_\omega-r_\omega/n}{\sigma_\omega}\right|\le\left|\frac{\xi_\omega-\mathbf{E}_Z\xi_\omega}{\widehat{\sigma}_\omega}\right|+\left|\frac{r_\omega^{(n)}-r_\omega}{n\widehat{\sigma}_\omega}\right|+|\mathbf{E}_Z\xi_\omega-r_\omega/n|\cdot\left|\frac{1}{\widehat{\sigma}_\omega}-\frac{1}{\sigma_\omega}\right|.$$

Since on $\Theta_c$, $\sigma_\omega\ge c$, according to Eq. (59), we can make $\widehat{\sigma}_\omega\ge c/2$ happen by assigning probability budget $\delta/2$ when $n$ is large enough. Also, combining Eq. (54), $|\mathbf{E}_Z\xi_\omega|\le 1$ and $r_\omega$ is a constant related to $\alpha$, then with $n$ large enough, there exist positive constants $C_5,C_6$,

$$\left|\frac{\xi_\omega-r_\omega^{(n)}/n}{\widehat{\sigma}_\omega}-\frac{\mathbf{E}_Z\xi_\omega-r_\omega/n}{\sigma_\omega}\right|\le\frac{2}{c}|\xi_\omega-\mathbf{E}_Z\xi_\omega|+\frac{C_5}{c}\sqrt{\frac{1}{n}\log\frac{1}{\alpha}}+\frac{C_6}{c^3}\left|\widehat{\sigma}_\omega^2-\sigma_\omega^2\right|. \tag{60}$$

Note that we need to pay for probability budget $\delta/2$ for the above conclusion to hold. Then by taking the supremum on both sides in Eq. (60) and assigning the remained probability budget $\delta/2$ to Eqs. (58) and (59), we can show that when $n$ is large enough, with probability at least $1-\delta$,

$$\sup_{(\theta_k,\theta_l)\in\Theta_c}\left|\frac{\xi_\omega-r_\omega^{(n)}/n}{\widehat{\sigma}_\omega}-\frac{\mathbf{E}_Z\xi_\omega-r_\omega/n}{\sigma_\omega}\right|$$
$$\sim\mathcal{O}\left(\frac{1}{c^3}\left[\sqrt{\frac{1}{n}\log\frac{1}{\delta}+(d_k+d_l)\frac{\log n}{n}}+\frac{R_{\omega_k}L_kR_{\Theta_k}+R_{\omega_l}L_lR_{\Theta_l}}{\sqrt{n}}\right]\right) \tag{61}$$

thus complete the proof. $\qquad\square$

# H    Proof of Theorem 3

In this section, we give a proof of the Theorem 3. We first restate the Theorem 3 here.

**Theorem 3** (Consistency)**.** *Let $\theta_k^*, \theta_l^*$ be the parameters after learning, $Z^{te}$ be the testing samples of size $m$, when $\mathbf{E}_Z HSIC_\omega^{(u)}(Z^{te}) > 0$, then the probability of Type II error*

$$\mathbb{P}\big(\text{Type II error}\big) = \mathbb{P}_{\mathcal{H}_1}\big(m HSIC_\omega(Z^{te}) \leq r_\omega^{(m)} | \theta_k^*, \theta_l^*\big) \sim \mathcal{O}(m^{-1/2}). \tag{62}$$

*Let the mapping functions with learned parameters $\theta_k^*, \theta_l^*$ be $\mathcal{T}_{\theta_k^*}, \mathcal{T}_{\theta_l^*}$, and the corresponding range space be a compact subset of $\mathbb{R}^{d_{\mathcal{T}_x}}, \mathbb{R}^{d_{\mathcal{T}_y}}$, respectively. Also, the diameter of two range spaces is denoted by $diam(\mathcal{T}_{\theta_k^*}), diam(\mathcal{T}_{\theta_l^*})$, respectively. Let $\{(\omega_{k;j}, \omega_{l;j})\}_{j=1}^{D/2}$ be the frequency samplings with their second moment denoted by $\sigma_{\omega_k}^2 := \mathbf{E}_{p_k(\omega)}[\omega_{k;j}^T \omega_{k;j}]$, $\sigma_{\omega_l}^2 := \mathbf{E}_{p_l(\omega)}[\omega_{l;j}^T \omega_{l;j}]$. Additionally, we denote $\xi_u := HSIC(X, Y)$, then under $\mathcal{H}_1$, we have $\mathbf{E}_Z HSIC_\omega^{(u)}(Z^{te}) > 0$ with any constant probability when $D = \Omega\Big(\frac{d_{\mathcal{T}_x} + d_{\mathcal{T}_y}}{\xi_u^2} \log \frac{\sigma_{\omega_k} diam(\mathcal{T}_{\theta_k^*}) + \sigma_{\omega_l} diam(\mathcal{T}_{\theta_l^*})}{\xi_u}\Big)$.*

*Proof.* The proof consists of two parts, we first give the rate of convergence of Type II error under condition $\mathbf{E}_Z HSIC_\omega^{(u)}(Z^{te}) > 0$, and next for condition $\mathbf{E}_Z HSIC_\omega^{(u)}(Z^{te}) > 0$ we give a lower bound on the number of frequency samplings required for it to hold. To simplify, we denote the U-statistic $HSIC_\omega^{(u)}(Z^{te})$ as $\xi_\omega^{(u)}$ in this proof. We begin to prove the first part. With the learned parameters $\theta_k^*, \theta_l^*$, the probability of the Type II error is given by

$$\mathbb{P}\big(\text{Type II error}\big) = \mathbb{P}_{\mathcal{H}_1}\Big(m\xi_\omega \leq r_\omega^{(m)} | \theta_k^*, \theta_l^*\Big). \tag{63}$$

Combing the result of the difference between $\xi_\omega$ and $\xi_\omega^{(u)}$ as shown in Eq. (34), we have

$$\mathbb{P}_{\mathcal{H}_1}\Big(m\xi_\omega \leq r_\omega^{(m)} | \theta_k^*, \theta_l^*\Big) \leq \mathbb{P}_{\mathcal{H}_1}\Big(m\xi_\omega^{(u)} \leq r_\omega^{(m)} + C_0 | \theta_k^*, \theta_l^*\Big), \tag{64}$$

where $C_0$ is a positive constant. To apply the rate of convergence of the Central Limit Theorem, we rewrite the right equation in Eq. (64) as

$$\mathbb{P}_{\mathcal{H}_1}\left(\frac{\sqrt{m}(\xi_\omega^{(u)} - \mathbf{E}_Z \xi_\omega^{(u)})}{4\sigma_\omega^{1/2}} \leq \frac{r_\omega^{(m)}/\sqrt{m} - \sqrt{m}\mathbf{E}_Z \xi_\omega^{(u)} + C_0/\sqrt{m}}{4\sigma_\omega^{1/2}} \Big| \theta_k^*, \theta_l^*\right), \tag{65}$$

where the standard deviation (defined in Proposition 1) $\sigma_\omega > 0$ under $\mathcal{H}_1$. Then according to the results in [34, Section 5.5.1 Theorem B], there exist nonnegative constant $C_1$ such that

$$\mathbb{P}\Big(\text{Type II error}\Big) \leq \Phi\left(\frac{r_\omega^{(m)}/\sqrt{m} - \sqrt{m}\mathbf{E}_Z \xi_\omega^{(u)} + C_0/\sqrt{m}}{4\sigma_\omega^{1/2}}\right) + \frac{C_1 \nu_h}{\sigma_\omega^{3/2}} \frac{1}{\sqrt{m}} \tag{66}$$

where $\nu_h := \mathbf{E}_Z^{i \neq j \neq q \neq r}|h_{ijqr}^{(\omega)}|^3 < \infty$. When $m$ is large enough, we further have

$$\mathbb{P}\Big(\text{Type II error}\Big) \leq \Phi\Big(C_2 - C_3\sqrt{m}\mathbf{E}_Z \xi_\omega^{(u)} + C_4/\sqrt{m}\Big) + C_5\frac{1}{\sqrt{m}}. \tag{67}$$

where $C_2, C_3, C_4$ are positive constants and using $r^{(m)} \sim \mathcal{O}(m^{1/2})$ we prove in Eq. (54). Hence when $\mathbf{E}_Z \xi_\omega^{(u)} > 0$, the leading term $\sqrt{m}\mathbf{E}_Z \xi_\omega^{(u)}$ decrease as $m$ increase. Further, to obtain the decrease rate when $m$ is close to infinity, we consider the asymptotic expansion (when $x$ is close to negative infinity) for the function $\Phi(x)$ as given by

$$\Phi(x) = -\frac{e^{-x^2}}{2x\sqrt{\pi}}\left(1 + \sum_{n=1}^{\infty}(-1)^n \frac{1 \cdot 3 \cdot 5 \cdots (2n-1)}{(2x^2)^n}\right), \tag{68}$$

thus $\Phi\Big(C_2 - C_3\sqrt{m}\mathbf{E}_Z \xi_\omega^{(u)} + C_4/\sqrt{m}\Big) \sim \mathcal{O}(m^{-1/2})$. As a result, the decreasing rate is at least $\mathcal{O}(m^{-1/2})$. We have so far completed the first part of the proof, and we next begin the second part of the proof, i.e. obtain the number of frequency samplings required for the condition $\mathbf{E}_Z \xi_\omega^{(u)} > 0$ to

hold. For simplify, we denote $\Lambda_k(x)^T \Lambda_k(x')$, $\Lambda_l(y)^T \Lambda_l(y')$ as $k^{(\omega)}(x, x')$, $l^{(\omega)}(y, y')$, respectively. Then according to Lemma 2, we have

$$\mathbb{P}\left[\sup_{x,x'\in\mathcal{X}} |k^{(\omega)}(x, x') - k(x, x')| \geq \epsilon\right] \leq 2^8 \left(\frac{\sigma_{\omega_k}\mathrm{diam}(\mathcal{T}_{\theta_k^*})}{\epsilon}\right)^2 \exp\left(-\frac{D\epsilon^2}{4(d_{\mathcal{T}_x}+2)}\right),$$

$$\mathbb{P}\left[\sup_{y,y'\in\mathcal{Y}} |l^{(\omega)}(y, y') - l(y, y')| \geq \epsilon\right] \leq 2^8 \left(\frac{\sigma_{\omega_l}\mathrm{diam}(\mathcal{T}_{\theta_l^*})}{\epsilon}\right)^2 \exp\left(-\frac{D\epsilon^2}{4(d_{\mathcal{T}_y}+2)}\right). \tag{69}$$

Also, we denote the bounds in Eq. (69) as $\delta_x(\epsilon, D), \delta_y(\epsilon, D)$, respectively. Next we get the bound between $\mathbf{E}_Z\xi_\omega^{(u)}$ and $\mathrm{HSIC}(X, Y)$. According to Lemma 3, the bound is given by

$$|\mathbf{E}_Z\xi_\omega^{(u)} - \mathrm{HSIC}(X, Y)| \leq 4 \cdot \sup_{x,x'\in\mathcal{X},y,y'\in\mathcal{Y}} |k^{(\omega)}(x, x')l^{(\omega)}(y, y') - k(x, x')l(y, y')|. \tag{70}$$

Since by the definition, for all $(x, x') \in \mathcal{X} \times \mathcal{X}, (y, y') \in \mathcal{Y} \times \mathcal{Y}$, we have $|k^{(\omega)}(x, x')| \leq 1$, $|l^{(\omega)}(y, y')| \leq 1, |k(x, x')| \leq 1, |l(y, y')| \leq 1$. Hence we have

$$|k^{(\omega)}(x, x')l^{(\omega)}(y, y') - k(x, x')l(y, y')| \leq |k^{(\omega)}(x, x') - k(x, x')| + |l^{(\omega)}(y, y') - l(y, y')|. \tag{71}$$

Combining the results of Eqs. (70) and (71), we obtain

$$|\mathbf{E}_Z\xi_\omega^{(u)} - \mathrm{HSIC}(X, Y)| \leq 4 \cdot \sup_{x,x'\in\mathcal{X}} |k^{(\omega)}(x, x') - k(x, x')| + 4 \cdot \sup_{y,y'\in\mathcal{Y}} |l^{(\omega)}(y, y') - l(y, y')|. \tag{72}$$

Combining the results as shown in Eq. (69) and allocating the probability budget $\epsilon$, we have

$$\mathbb{P}\left[\sup_{x,x'\in\mathcal{X},y,y'\in\mathcal{Y}} |\mathbf{E}_Z\xi_\omega^{(u)} - \mathrm{HSIC}(X, Y)| \geq \epsilon\right] \leq \delta_x(\epsilon/8, D) + \delta_y(\epsilon/8, D). \tag{73}$$

By setting $\epsilon = \xi_u/2$, and since $\xi_u > 0$ under $\mathcal{H}_1$, we conclude that $\mathbf{E}_Z\xi_\omega^{(u)} > 0$ holds with any constant probability when $D = \Omega\left(\frac{d_{\mathcal{T}_x}+d_{\mathcal{T}_y}}{\xi_u^2} \log \frac{\sigma_{\omega_k}\mathrm{diam}(\mathcal{T}_{\theta_k^*})+\sigma_{\omega_l}\mathrm{diam}(\mathcal{T}_{\theta_l^*})}{\xi_u}\right)$. $\qquad\square$

In the proof of Theorem 3, we obtain the convergence bound of the statistic $\mathbf{E}_Z\xi_\omega^{(u)}$. Actually, the convergence result for its estimation can also be obtained, as shown in the following Corollary.

**Corollary 1** (Approximation Error Bound of $\mathrm{HSIC}_\omega(Z^{te})$). *Maintaining the same conditions and notions as in Theorem 3, we have the uniform convergence bound of $\mathrm{HSIC}_\omega(Z^{te})$ as*

$$\mathbb{P}\left[\sup_{Z^{te}\in\mathcal{X}\times\mathcal{Y}} |\mathrm{HSIC}_\omega(Z^{te}) - \mathrm{HSIC}_b(Z^{te})| \geq \epsilon\right] \leq \delta_x(\epsilon/8, D) + \delta_y(\epsilon/8, D). \tag{74}$$

*Proof.* By the definition, we have for all $i, j, q, r$, $|k_{ij}^{(\omega)}| \leq 1, |l_{ij}^{(\omega)}| \leq 1, |k_{ij}| \leq 1, |l_{ij}| \leq 1$, thus

$$|k_{ij}^{(\omega)}l_{qr}^{(\omega)} - k_{ij}l_{qr}| \leq |k_{ij}^{(\omega)} - k_{ij}||l_{qr}^{(\omega)}| + |k_{ij}||l_{qr}^{(\omega)} - l_{qr}| \leq |k_{ij}^{(\omega)} - k_{ij}| + |l_{qr}^{(\omega)} - l_{qr}|. \tag{75}$$

Then according to the results as shown in Eq. (69), we have for all $i, j, q, r$

$$\mathbb{P}\left[\sup_{x_i,x_j\in\mathcal{X},y_q,y_r\in\mathcal{Y}} |k_{ij}^{(\omega)}l_{qr}^{(\omega)} - k_{ij}l_{qr}| \geq \epsilon\right] \leq \delta_x(\epsilon/2, D) + \delta_y(\epsilon/2, D). \tag{76}$$

Recall the definition of $h_{ijqr}^{(\omega)} = \frac{1}{4!}\sum_{(t,u,v,w)}^{(i,j,q,r)} k_{tu}^{(\omega)}l_{tu}^{(\omega)} + k_{tu}^{(\omega)}l_{vw}^{(\omega)} - 2k_{uv}^{(\omega)}l_{tv}^{(\omega)}$ and we further define the corresponding $h_{ijqr} = \frac{1}{4!}\sum_{(t,u,v,w)}^{(i,j,q,r)} k_{tu}l_{tu} + k_{tu}l_{vw} - 2k_{uv}l_{tv}$, then for all $i, j, q, r$,

$$\mathbb{P}\left[\sup_{x_i,x_j\in\mathcal{X},y_q,y_r\in\mathcal{Y}} |h_{ijqr}^{(\omega)} - h_{ijqr}| \geq \epsilon\right] \leq \delta_x(\epsilon/8, D) + \delta_y(\epsilon/8, D). \tag{77}$$

After that, using the expressions that we obtained before, i.e., $\mathrm{HSIC}_\omega(Z^{te}) := \frac{1}{n^4}\sum_{i,j,q,r} h_{ijqr}^{(\omega)}$ and $\mathrm{HSIC}_b(Z^{te}) := \frac{1}{n^4}\sum_{i,j,q,r} h_{ijqr}$, we obtain the final bound that

$$\mathbb{P}\left[\sup_{Z^{te}\in\mathcal{X}\times\mathcal{Y}} |\mathrm{HSIC}_\omega(Z^{te}) - \mathrm{HSIC}_b(Z^{te})| \geq \epsilon\right] \leq \delta_x(\epsilon/8, D) + \delta_y(\epsilon/8, D). \tag{78}$$

and thus complete the proof. $\qquad\square$

# I Smoothness of Optimization Objective

We first prove the Lipschitz property for some functions. For ease of reference, we re-list here the definitions of the terms that related to the optimization objective: $\xi_\omega := \mathrm{HSIC}_\omega(Z) = \frac{1}{n^4} \sum_{i,j,q,r} h_{ijqr}^{(\omega)}$, $\widehat{\sigma}_\omega^2 := 16\left[\frac{1}{n} \sum_i (\frac{1}{n^3} \sum_{j,q,r} h_{ijqr}^{(\omega)})^2 - \mathrm{HSIC}_\omega^2(Z)\right]$ and $\sigma_\omega^2 := 16\left[\mathbf{E}_i(\mathbf{E}_{j,q,r} h_{ijqr}^{(\omega)})^2 - (\mathbf{E}_Z h_{ijqr}^{(\omega)})^2\right]$. The Lipschitz property of these terms are shown as follows.

**Lemma 4** (Lipschitz Property of of $\xi_\omega, \mathbf{E}_Z \xi_\omega, \widehat{\sigma}_\omega^2, \sigma_\omega^2$). *Maintaining the same conditions and notions as in Lemma 1, we have the following Lipschitz property*

$$
\begin{aligned}
|\xi_\omega(\theta_k, \theta_l) - \xi_\omega(\theta_k', \theta_l')| &\leq 8R_{\omega_k} L_k \cdot \|\theta_k - \theta_k'\| + 8R_{\omega_l} L_l \cdot \|\theta_l - \theta_l'\|, \\
|\mathbf{E}_Z[\xi_\omega(\theta_k, \theta_l)] - \mathbf{E}_Z[\xi_\omega(\theta_k', \theta_l')]| &\leq 8R_{\omega_k} L_k \cdot \|\theta_k - \theta_k'\| + 8R_{\omega_l} L_l \cdot \|\theta_l - \theta_l'\|, \\
|\widehat{\sigma}_\omega^2(\theta_k, \theta_l) - \widehat{\sigma}_\omega^2(\theta_k', \theta_l')| &\leq 1024R_{\omega_k} L_k \cdot \|\theta_k - \theta_k'\| + 1024R_{\omega_l} L_l \cdot \|\theta_l - \theta_l'\|, \\
|\sigma_\omega^2(\theta_k, \theta_l) - \sigma_\omega^2(\theta_k', \theta_l')| &\leq 1024R_{\omega_k} L_k \cdot \|\theta_k - \theta_k'\| + 1024R_{\omega_l} L_l \cdot \|\theta_l - \theta_l'\|,
\end{aligned}
\tag{79}
$$

*where we use the symbol $\xi_\omega(\theta_k, \theta_l)$ to denote $\xi_\omega$ with the parameter $\theta_k, \theta_l$ and the others by analogy.*

*Proof.* We start by obtaining the result of $h_{ijqr}^{(\omega)}$ for all $i, j, q, r$. Since for all $i, j, q, r$,

$$
\left|k_{ij}^{(\omega)}(\theta_k) l_{qr}^{(\omega)}(\theta_l) - k_{ij}^{(\omega)}(\theta_k') l_{qr}^{(\omega)}(\theta_l')\right| \leq \left|k_{ij}^{(\omega)}(\theta_k) - k_{ij}^{(\omega)}(\theta_k')\right| + \left|l_{qr}^{(\omega)}(\theta_l) - l_{qr}^{(\omega)}(\theta_l')\right|,
\tag{80}
$$

where the property $|k_{ij}^{(\omega)}| \leq 1$ and $|l_{qr}^{(\omega)}| \leq 1$ are used. According the Lemma 1, we have

$$
\left|k_{ij}^{(\omega)}(\theta_k) - k_{ij}^{(\omega)}(\theta_k')\right| \leq 2R_{\omega_k} L_k \cdot \|\theta_k - \theta_k'\|, \quad \left|l_{qr}^{(\omega)}(\theta_l) - l_{qr}^{(\omega)}(\theta_l')\right| \leq 2R_{\omega_l} L_l \cdot \|\theta_l - \theta_l'\|.
\tag{81}
$$

Combing the definition that $h_{ijqr}^{(\omega)} := \frac{1}{4!} \sum_{(t,u,v,w)}^{(i,j,q,r)} k_{tu}^{(\omega)} l_{tu}^{(\omega)} + k_{tu}^{(\omega)} l_{vw}^{(\omega)} - 2k_{uv}^{(\omega)} l_{tv}^{(\omega)}$, then

$$
\left|h_{ijqr}^{(\omega)}(\theta_k, \theta_l) - h_{ijqr}^{(\omega)}(\theta_k', \theta_l')\right| \leq 8R_{\omega_k} L_k \cdot \|\theta_k - \theta_k'\| + 8R_{\omega_l} L_l \cdot \|\theta_l - \theta_l'\|.
\tag{82}
$$

Also, combining the definition of $\xi_\omega := \mathrm{HSIC}_\omega(Z) = \frac{1}{n^4} \sum_{i,j,q,r} h_{ijqr}^{(\omega)}$, we obtain

$$
|\xi_\omega(\theta_k, \theta_l) - \xi_\omega(\theta_k', \theta_l')| \leq 8R_{\omega_k} L_k \cdot \|\theta_k - \theta_k'\| + 8R_{\omega_l} L_l \cdot \|\theta_l - \theta_l'\|.
\tag{83}
$$

By using $|\mathbf{E}_Z[\xi_\omega(\theta_k, \theta_l)] - \mathbf{E}_Z[\xi_\omega(\theta_k', \theta_l')]| \leq \mathbf{E}_Z[|\xi_\omega(\theta_k, \theta_l) - \xi_\omega(\theta_k', \theta_l')|]$, we have

$$
|\mathbf{E}_Z[\xi_\omega(\theta_k, \theta_l)] - \mathbf{E}_Z[\xi_\omega(\theta_k', \theta_l')]| \leq 8R_{\omega_k} L_k \cdot \|\theta_k - \theta_k'\| + 8R_{\omega_l} L_l \cdot \|\theta_l - \theta_l'\|.
\tag{84}
$$

For the results of $\widehat{\sigma}_\omega^2, \sigma_\omega^2$, we first proof the following results. For all $i, j, q, r, i', j', q', r'$, we have

$$
\begin{aligned}
&\left|h_{ijqr}^{(\omega)}(\theta_k, \theta_l) h_{i'j'q'r'}^{(\omega)}(\theta_k, \theta_l) - h_{ijqr}^{(\omega)}(\theta_k', \theta_l') h_{i'j'q'r'}^{(\omega)}(\theta_k', \theta_l')\right| \\
\leq \ & 4 \cdot \left|h_{ijqr}^{(\omega)}(\theta_k, \theta_l) - h_{ijqr}^{(\omega)}(\theta_k', \theta_l')\right| + 4 \cdot \left|h_{i'j'q'r'}^{(\omega)}(\theta_k, \theta_l) - h_{i'j'q'r'}^{(\omega)}(\theta_k', \theta_l')\right| \\
\leq \ & 64R_{\omega_k} L_k \cdot \|\theta_k - \theta_k'\| + 64R_{\omega_l} L_l \cdot \|\theta_l - \theta_l'\|,
\end{aligned}
\tag{85}
$$

where the first inequality holds due to property $|h_{ijqr}^{(\omega)}| \leq 4$. Then we use the expression

$$
\widehat{\sigma}_\omega^2 = 16\left[\frac{1}{n^7} \sum_{i,j,q,r,j',q',r'} h_{ijqr}^{(\omega)} h_{ij'q'r'}^{(\omega)} - \frac{1}{n^8} \sum_{i,j,q,r,i',j',q',r'} h_{ijqr}^{(\omega)} h_{i'j'q'r'}^{(\omega)}\right]
\tag{86}
$$

and combine the results in Eq. (85). As a result, we obtain that

$$
|\widehat{\sigma}_\omega^2(\theta_k, \theta_l) - \widehat{\sigma}_\omega^2(\theta_k', \theta_l')| \leq 1024R_{\omega_k} L_k \cdot \|\theta_k - \theta_k'\| + 1024R_{\omega_l} L_l \cdot \|\theta_l - \theta_l'\|.
\tag{87}
$$

In a similar way, we can obtain the corresponding expression of $\sigma_\omega^2$ as

$$
\sigma_\omega^2 = 16\left[\mathbf{E}_{i,j,q,r,j',q',r'} h_{ijqr}^{(\omega)} h_{ij'q'r'}^{(\omega)} - \mathbf{E}_{i,j,q,r,i',j',q',r'} h_{ijqr}^{(\omega)} h_{i'j'q'r'}^{(\omega)}\right].
\tag{88}
$$

Then we obtain a similar result as before, i.e.,

$$
|\sigma_\omega^2(\theta_k, \theta_l) - \sigma_\omega^2(\theta_k', \theta_l')| \leq 1024R_{\omega_k} L_k \cdot \|\theta_k - \theta_k'\| + 1024R_{\omega_l} L_l \cdot \|\theta_l - \theta_l'\|
\tag{89}
$$

which completes the proof. $\qquad\square$

Also for the term associated with the estimated threshold (recall that it is computed from the first two moments), we obtain the following properties of $\mathcal{E}_0, \mathcal{V}_0$ as defined in Theorem 1.

**Lemma 5** (Lipschitz Property of of $\mathcal{E}_0, \mathcal{V}_0$). *Maintaining the same conditions and notions as in Lemma 4, we have the following Lipschitz property*

$$
\begin{aligned}
|\mathcal{E}_0(\theta_k, \theta_l) - \mathcal{E}_0(\theta_k', \theta_l')| &\leq 2C_0 R_{\omega_k} L_k \cdot \|\theta_k - \theta_k'\| + 2C_0 R_{\omega_l} L_l \cdot \|\theta_l - \theta_l'\|, \\
|\mathcal{V}_0(\theta_k, \theta_l) - \mathcal{V}_0(\theta_k', \theta_l')| &\leq 128 C_1 R_{\omega_k} L_k \cdot \|\theta_k - \theta_k'\| + 128 C_1 R_{\omega_l} L_l \cdot \|\theta_l - \theta_l'\|,
\end{aligned}
\tag{90}
$$

*where the constant $C_0(n) := \frac{n^2}{(n-1)^2}$ and $C_1(n) := \frac{n(n-4)(n-5)}{(n-1)(n-2)(n-3)}$.*

*Proof.* The expression in Theorem 1, while easy to compute, is not suitable for this part of our proof. We begin by obtaining equivalent expressions for $\mathcal{E}_0, \mathcal{V}_0$. According to Eq. (39), we have

$$
\begin{aligned}
\mathcal{E}_0 &= \left[ 1 - \frac{\mathbf{1}^T(\mathbf{\Lambda}_X \mathbf{\Lambda}_X^T - \mathbf{I}_n)\mathbf{1}}{n(n-1)} \right] \left[ 1 - \frac{\mathbf{1}^T(\mathbf{\Lambda}_Y \mathbf{\Lambda}_Y^T - \mathbf{I}_n)\mathbf{1}}{n(n-1)} \right] \\
&= C_0 \left[ 1 - \frac{\mathbf{1}^T(\mathbf{\Lambda}_X \mathbf{\Lambda}_X^T)\mathbf{1}}{n^2} \right] \left[ 1 - \frac{\mathbf{1}^T(\mathbf{\Lambda}_Y \mathbf{\Lambda}_Y^T)\mathbf{1}}{n^2} \right].
\end{aligned}
\tag{91}
$$

And for $\mathcal{V}_0$, we use Eq. (45) and obtain

$$
\mathcal{V}_0 = \frac{2C_1}{n^4} \left[ \mathrm{Tr}(\mathbf{\Lambda}_X \mathbf{\Lambda}_X^T \mathbf{H} \mathbf{\Lambda}_X \mathbf{\Lambda}_X^T \mathbf{H}) \right] \left[ \mathrm{Tr}(\mathbf{\Lambda}_Y \mathbf{\Lambda}_Y^T \mathbf{H} \mathbf{\Lambda}_Y \mathbf{\Lambda}_Y^T \mathbf{H}) \right].
\tag{92}
$$

Also, we define $h_{ijqr}^{(k)} := \frac{1}{4!} \sum_{(t,u,v,w)}^{(i,j,q,r)} k_{tu}^{(\omega)} k_{tu}^{(\omega)} + k_{tu}^{(\omega)} k_{vw}^{(\omega)} - 2 k_{uv}^{(\omega)} k_{tv}^{(\omega)}$ that corresponding to $h_{ijqr}^{(\omega)}$ and also define $h_{ijqr}^{(l)} := \frac{1}{4!} \sum_{(t,u,v,w)}^{(i,j,q,r)} l_{tu}^{(\omega)} l_{tu}^{(\omega)} + l_{tu}^{(\omega)} l_{vw}^{(\omega)} - 2 l_{uv}^{(\omega)} l_{tv}^{(\omega)}$. Then we can further rewrite the term for $X$ as $\frac{1}{n^2} \mathrm{Tr}(\mathbf{\Lambda}_X \mathbf{\Lambda}_X^T \mathbf{H} \mathbf{\Lambda}_X \mathbf{\Lambda}_X^T \mathbf{H}) = \frac{1}{n^4} \sum_{i,j,q,r} h_{ijqr}^{(k)}$ and for $Y$ by analogy. Then the properties of $h_{ijqr}^{(\omega)}$ can also be obtained for $h_{ijqr}^{(k)}$ and $h_{ijqr}^{(l)}$, e.g., $|h_{ijqr}^{(k)}| \leq 4, |h_{ijqr}^{(l)}| \leq 4$ for all $i, j, q, r$. Next we start to prove the Lipschitz property of $\mathcal{E}_0, \mathcal{V}_0$. For $\mathcal{E}_0$, according to Eq. (81) and combining the results $0 \leq \mathbf{1}^T(\mathbf{\Lambda}_X \mathbf{\Lambda}_X^T)\mathbf{1} \leq n^2$ and that for $Y$, we can show that

$$
\begin{aligned}
|\mathcal{E}_0(\theta_k, \theta_l) - \mathcal{E}_0(\theta_k', \theta_l')| &\leq \frac{C_0}{n^2} \sum_{i,j} |k_{ij}^{(\omega)}(\theta_k) - k_{ij}^{(\omega)}(\theta_k')| + \frac{C_0}{n^2} \sum_{q,r} |l_{qr}^{(\omega)}(\theta_l) - l_{qr}^{(\omega)}(\theta_l')| \\
&\leq 2C_0 R_{\omega_k} L_k \cdot \|\theta_k - \theta_k'\| + 2C_0 R_{\omega_l} L_l \cdot \|\theta_l - \theta_l'\|,
\end{aligned}
\tag{93}
$$

where $\mathbf{1}^T(\mathbf{\Lambda}_X \mathbf{\Lambda}_X^T)\mathbf{1} = \sum_{i,j} k_{ij}^{(\omega)}$ by definition and so as for $Y$. And for $\mathcal{V}_0$, we can prove that

$$
\begin{aligned}
&|\mathcal{V}_0(\theta_k, \theta_l) - \mathcal{V}_0(\theta_k', \theta_l')| \\
&\leq \frac{2C_1}{n^8} \sum_{i,j,q,r,i',j',q',r'} |h_{ijqr}^{(k)}(\theta_k, \theta_l) h_{i'j'q'r'}^{(l)}(\theta_k, \theta_l) - h_{ijqr}^{(k)}(\theta_k', \theta_l') h_{i'j'q'r'}^{(l)}(\theta_k', \theta_l')| \\
&\leq 128 C_1 R_{\omega_k} L_k \cdot \|\theta_k - \theta_k'\| + 128 C_1 R_{\omega_l} L_l \cdot \|\theta_l - \theta_l'\|
\end{aligned}
\tag{94}
$$

where the last inequation is obtained similar to Eq. (85), thus completes the proof. $\square$

The following results extend the results in [30] to the more general case (we only restrict the mapping functions to satisfy the Lipschitz property, and thus include the Gaussian kernel case of their proof).

**Theorem 4** (Smoothness of Optimization Objective). *Let the sample of size $n$ be $Z$, and with a small positive constant $c$, let the set of the parameters be $\overline{\Theta}_c := \{(\theta_k, \theta_l) | \widehat{\sigma}_\omega \geq c, \mathcal{V}_0 \geq c, \mathcal{E}_0 \geq c\}$, then there exist a nonnegative constant $L$ such that $\|\nabla_{(\theta_k, \theta_l)} J\| \leq L$ on $\overline{\Theta}_c$, where the optimization objective $J := [HSIC_\omega(Z) - \widehat{c_\alpha}/n]/\widehat{\sigma}_\omega$ is that we used in practice.*

*Proof.* According to Lemma 4, we have shown that $\mathrm{HSIC}_\omega(Z), \widehat{\sigma}_\omega$ both fits the Lipschitz condition. Also, according to Lemma 5, we have shown that $\mathcal{E}_0, \mathcal{V}_0$ are also Lipschitz with respect to $\theta_k, \theta_l$. Since the threshold $\widehat{c_\alpha}$ is completely determined by these two moments, combining the smoothness property of the mapping from two moments to thresholds as analyzed in [30, Theorem 2], we obtain that $\widehat{c_\alpha}$ is also Lipschitz with respect to $\theta_k, \theta_l$ on $\overline{\Theta}_c$. As a result, we complete the entire proof based on the Lipschitz property of composite mappings. $\square$

**Remark.** $\mathcal{E}_0$ and $\mathcal{V}_0$ are positive is almost satisfied in practice since according to the definition only $[\mathbf{1}^T \mathbf{\Lambda}_{Xc}^{\cdot 2} \mathbf{1}][\mathbf{1}^T \mathbf{\Lambda}_{Yc}^{\cdot 2} \mathbf{1}]$ and $[\mathbf{1}^T(\mathbf{\Lambda}_{Xc}^T \mathbf{\Lambda}_{Xc})^{\cdot 2} \mathbf{1}][\mathbf{1}^T(\mathbf{\Lambda}_{Yc}^T \mathbf{\Lambda}_{Yc})^{\cdot 2} \mathbf{1}]$ need to be greater than 0.

# J  Details of Experiment Setup

In this section, we give an introduction to the comparison methods in our experiments and provide the implementation details of each method.

## J.1  Details of Comparison Methods

The methods of comparison used in the experiment are described below.

- dCor [39]: An independence test that is based on the distance covariance.
- QHSIC [14]: The original quadratic-time HSIC independence test.
- RDC [26]: The randomized dependence coefficient that measures the independence using the canonical correlation between a finite set of random features of the copula.
- NyHSIC [44]: A variant of HSIC that uses the Nyström method to approximate kernels.
- FHSIC [44]: A variant of HSIC that uses the random Fourier feature to approximate kernels.
- BHSIC [44]: A variant of HSIC with the block-based statistic.
- HSICAgg [32]: An aggregated kernel test with the incomplete statistic of HSIC.
- NFSIC [18]: A test uses the normalized version of the finite set independence criterion and chooses features on a hold-out validation set to optimize a lower bound on the test power.

Below are the **GitHub URLs** for each comparison method.

- dCor: `https://pypi.org/project/dcor`.
- QHSIC: `https://github.com/amber0309/HSIC/blob/master/HSIC.py`.
- RDC: `https://github.com/lopezpaz/randomized_dependence_coefficient`.
- NyHSIC: `https://github.com/oxcsml/kerpy/blob/master/independence_testing`.
- FHSIC: `https://github.com/oxcsml/kerpy/blob/master/independence_testing`.
- BHSIC: `https://github.com/oxcsml/kerpy/blob/master/independence_testing`.
- HSICAgg: `https://github.com/antoninschrab/mmdagg/tree/master/mmdagg`.
- NFSIC: `https://github.com/wittawatj/fsic-test/blob/master/fsic`.

**Time Complexity.** Among them, dCor and QHSIC are the tests of quadratic complexity with sample size $n$, i.e., $\mathcal{O}(n^2)$. RDC is calculated in $\mathcal{O}(n \log n)$ and the rest are linear-time tests, i.e., $\mathcal{O}(n)$.

**Threshold.** For dCor, QHSIC, RDC, NyHSIC, FHSIC, and BHSIC, we permute the samples 100 times to simulate the null distribution and compute the threshold. The thresholds for the remaining methods are obtained by asymptotic null distribution, i.e., we set the test threshold to the $(1 - \alpha)$-quantile of $\chi^2(J)$ for NFSIC and obtain the test threshold of LFHSIC-G/M by gamma approximation.

**Details of Setup.** The number of random features for FHSIC, LFHSIC-G/M, the number of induced variables for NyHSIC, the block size for BHSIC and the number of sub-diagonals $R$ for HSICAgg are all kept consistent as recommended in [44] for fair evaluation. Specifically, we set the number of random mappings in RDC to 20 to ensure compatibility with large-scale datasets. The test location parameter $J$ of NFHSIC is set as default as 10, since it differs from other approximation methods that blindly increasing $J$ may lead to a loss of power as shown in [18] and can significantly escalate time costs due to its cubic time complexity $\mathcal{O}(J^3)$. In the optimization step, for stabilizing the training, in the implementation of NFSIC we determine the initial bandwidth by searching the best from 25 bandwidth combinations (including the median bandwidth combination). For LFHSIC-G/M, to be fair, we perform the same grid search on SD, Sin, and GSign datasets. In other experiments, we still use the median bandwidth as initialization for LFHSIC-G/M. Also, the maximum number of iterations for the optimization is set to 100 for NFSIC and LFHSIC-G/M. The default learning rate of the optimization of LFHSIC-G/M is set as 0.05 in all the experiments. As for HSICAgg, the default implementation of the predefined 25 pairs of bandwidths in its code is used. For synthetic data, we set the split ratio to 0.5 for NFSIC and LFHSIC-G/M, i.e., we randomly sample half of the data for training and use the remaining for independence testing, while the other methods use all data for testing. For real MSD data, we divide a small portion of the data for training and then extract 100 random subsets of the remaining data (disjoint from the training set) for evaluation.

## J.2 Details of Datasets

The details of the four synthetic datasets and two real datasets are described below.

- **Sine Dependency (SD):** In this model, $X$ follows a $d$-dimension multivariate normal distribution $\mathcal{N}_d(0, I_d)$, and $Y$ is defined as $20\sin\left(4\pi(X_1^2 + X_2^2)\right) + Z$, where $X_i$ is the $i$-th dimension of $X$, and $Z \sim \mathcal{N}(0, 1)$ represents independent noise. Notably, when $d > 2$, $Y$ exhibits a nonlinear relationship solely with the first two dimensions of $X$.

- **Sinusoid (Sin):** This model introduces a localized alteration in the probability density function $p_{xy}$ over $\mathcal{X} \times \mathcal{Y} := [-\pi, \pi]^2$, specified as $(X, Y) \sim p_{xy}(x, y) \propto 1 + \sin(\omega x)\sin(\omega y)$, where $\omega$ denotes the frequency. Increasing the frequency enhances the similarity between the sampled data and that drawn from $\text{Uniform}([-\pi, \pi]^2)$, thereby augmenting the challenge of detecting dependency with limited sample sizes. An example visualization is shown on the left of Fig. 6.

- **Gaussian Sign (GSign):** In this model, $X$ follows a $d$-dimension multivariate normal distribution $\mathcal{N}_d(0, I_d)$, and $Y$ is expressed as $|Z| \prod_{i=1}^d \text{sgn}(X_i)$, where $\text{sgn}(\cdot)$ represents the sign function, $X_i$ denotes the $i$-th dimension of $X$, and $Z \sim \mathcal{N}(0, 1)$ is independent of $X$. The challenge lies in $Y$ being independent of any proper subset of $X$ but dependent on $X$ as a whole, underscoring the importance of considering all dimensions of $X$ simultaneously in independence testing.

- **ISA Dataset.** We construct the data through the following steps: First, we generate $n$ i.i.d samples of two univariate random variables with a mixture of Gaussian model, i.e. $\frac{1}{2}\mathcal{N}(-1, 0.01) + \frac{1}{2}\mathcal{N}(1, 0.01)$. Second, we mix these random variables using a rotation matrix parameterized by an angle $\theta$, which varies from 0 to $\pi/4$. A zero angle implies independence between the data, while a larger angle signifies stronger dependency. Third, we append noise with a distribution of $\mathcal{N}_{d-1}(0, I_{d-1})$ to each of the mixtures. Finally, we multiply an independent random $d$-dimensional orthogonal matrix to obtain vectors dependent across all observed dimensions. The resulting random variables $X$ and $Y$ are dependent but uncorrelated. When $d$ is greater than 1, the problem is associated with the independent subspace analysis (ISA) problem [14]. For the case $d = 1, \theta = \pi/10$, an example visualization is shown in the middle of Fig. 6.

- **3DShapes Dataset.** This dataset [5] comprises images depicting 3D scenes, complete with additional features like shadows and backgrounds. It encompasses six fundamental latent factors: floor hue, wall hue, object hue, object scale, object shape, and orientation, all adjustable to generate corresponding images. Orientation is treated as a dependency factor for independence testing, where we test the dependency between the image $X$ and its orientation $Y$. To heighten the challenge, we maintain the object shape as a ball, minimizing the apparent orientation feature compared to other shapes like squares, while randomizing the remaining factors. An example visualization is given on the right side of Fig. 6.

- **Million Song Dataset.** The dataset, a subset of the Million Song Data[3] [4], comprises $515,345$ songs with 91-dimensional features. The first dimension represents the release year of each song, designated as variable $Y$, while the remaining features (e.g., timbre average and timbre covariance) form variable $X$. Our objective is to identify the dependency between $X$ and $Y$.

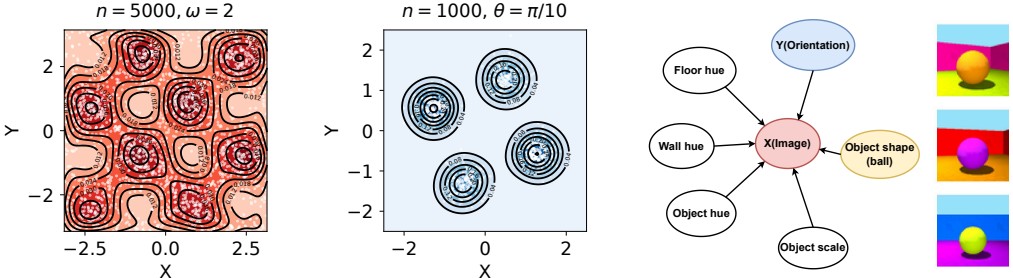

Figure 6: Examples of visualization of samples from different datasets. The two plots on the left correspond to the samples and their contour under Sin dataset ($n = 5000, w = 2$), and ISA dataset ($n = 1000, d = 1, \theta = \pi/10$), respectively. Right: a visualization of the causal diagram of the data generation process and some generated examples.

---

[3]Million Song Data subset: `https://archive.ics.uci.edu/dataset/203/yearpredictionmsd`

## K  Additional Experiment Results

In this section, we provide additional experimental results, mainly including the visualization results on the Sin synthetic dataset, the results with more comparing methods (as explained in the main paper, due to the high time overhead therefore do not participate in the evaluation of the main paper) as well as the running time of each method.

### K.1  The visualization results on the Sin model

We provide the visualization results on the Sin synthetic dataset to illustrate the performance of our optimization objective for the example mentioned in the main paper. The results are shown in Fig. 7, where the setup follows our experiments ($n = 2000, \omega = 5, D = 100$). For visualization, the negative of our optimization objective $J$ is shown. As can be seen, our optimization objective guides to letting the bandwidth adapt to improve the test power, and here we can see that regions with bandwidths around $0.2$ (corresponding to the theoretical optimal solution in our main paper) indicate better power, thus corroborating the validity of our optimization objective. Also, notice that the landscape is smooth over a wide range as demonstrated in Theorem 4, and thus contributes to non-convex optimization.

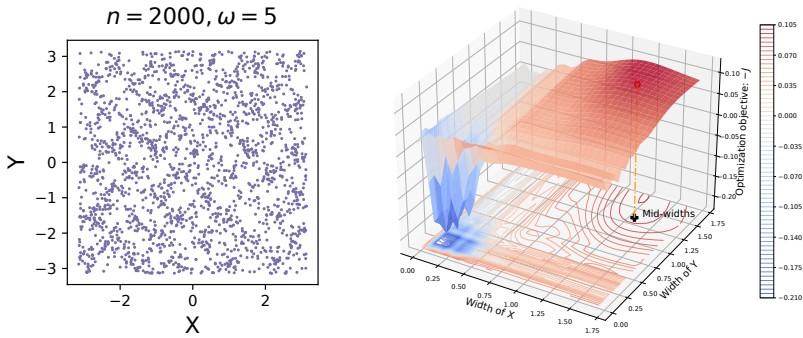

Figure 7: The visualization results on the Sin model. Left: the samples with $n = 2000, \omega = 5$. The visualization of the landscape of the negative of our optimization objective ($D = 100$).

### K.2  Additional experimental comparisons

As mentioned in the main paper, the method[4] [30] is not involved in the comparison because it takes too much time to run in some settings. Here, we compare it with ours under some feasible settings to illustrate the improvement of our method on power-runtime trade-offs. The methods using Gaussian kernel with global width and Gaussian kernel with widths of each dimension (corresponding with ours) are employed, referred to QHSIC-O and QHSIC-W, respectively. For fairness, the same grid search procedure is employed as the initialization of the optimization. We perform the evaluation on the SD data and plot the results of the test power over time as shown in Fig. 8. Also, for our methods, we provide the results under the setting $D = 100$ and $D = 500$ as in the main paper. The experiments are conducted with the same equipment, specifically a 6-core CPU with a 3080 GPU.

**Results.** Our test consistently results in a better power-runtime tradeoff at different $D$ settings. At $D = 100$, one test can be completed in less than a second when the sample size $n = 6000$. As $D$ increases ($D = 500$), the number of samples required to achieve the same power decreases, but the increase in $D$ leads to an overhead in runtime, and overall our test is still completed in a few seconds. In contrast, even though QHSIC-O/W requires fewer samples to reach the same power than our tests, the runtime rises rapidly as the sample size increases. When $n = 1000$, it already takes more than $10s$ to perform a test. When $n = 3000$, it needs nearly a minute to perform a test, which may greatly limit the practical application.

---

[4]The code is downloaded from `https://github.com/renyixin666/HSIC-LK`

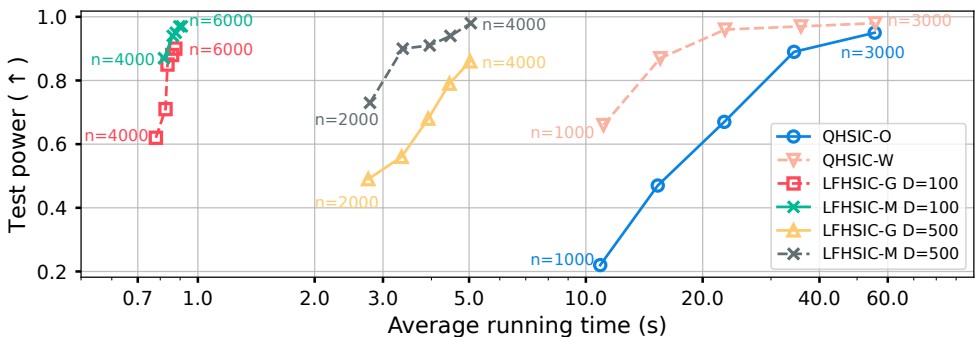

Figure 8: Time-power trade-off curves on the SD dataset.

### K.3 Running Time

In this part, we evaluate the running time of each method on ISA datasets $d = 10$. We set $D = 100$ and plot the results of the running time versus sample size $n$ as shown in Fig. 9. Shown on the left are the results of tests with $O(n)$ and $O(n \log n)$ time complexity. Since the quadratic time complexity test cannot handle large-scale inputs of 100,000 sample size (excessive runtime and large memory overhead to store the kernel matrix), we evaluate them separately on the right. The experiments are all conducted on the same equipment, specifically a 14-core CPU with a 4090 GPU.

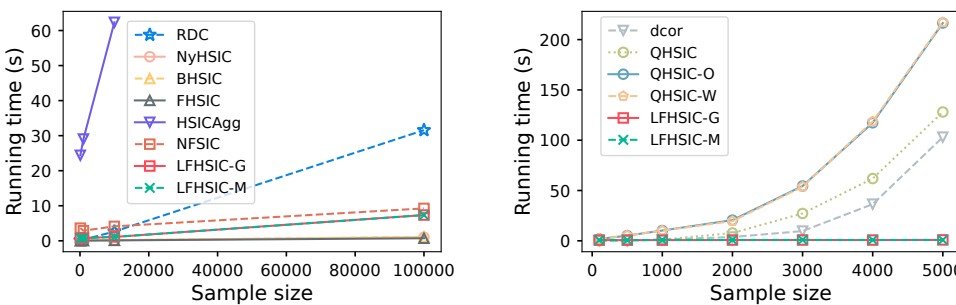

Figure 9: The running time curves with sample size $n$ on the ISA dataset ($d = 10$).

**Results.** The experimental results on the left show that our method is faster than other methods with optimizable options (HSICAgg and NFSIC), and can complete a test within 10 seconds even with $100,000$ samples. Even though HSICAgg uses parallelism to optimize the computational efficiency of the scheme, the actual implementation of the parallelism is time-consuming, and hence leads to a high time overhead for a single practical test. For the results on the right, it can be seen that the quadratic complexity methods face a dramatic increase in time overhead as the sample size rises, and this is especially severe for the methods (QHSIC-O/W) that need to be optimized since the optimizing objective needs to perform multiple squared complexity operations. In contrast, our linear-time learning objective allows us to handle huge data samples very efficiently.

## L   Limitations and Broader Impacts

**Limitations.** According to the experimental results in the main paper as well as in the Appendix, no one method is better than the others in all settings, so it is important to choose several appropriate tests for real scenarios and summarize their results in order to obtain a more reliable conclusion.

**Broader Impacts.** This work proposes a novel framework for independence testing. The proposed linear-time optimization objective can be trained end-to-end in a data-driven manner, ensuring both effectiveness and efficiency in high-dimensional and large-scale scenarios. This could be beneficial for developing more reliable downstream algorithms in a variety of areas, including causal discovery, feature selection, and deep learning.

