# OpenReview forum: "Efficiently Learning Significant Fourier Feature Pairs for Statistical Independence Testing"
_NeurIPS.cc/2024/Conference — NeurIPS 2024 poster_

### Official Review · Reviewer_uRf4 · 2024-07-12

**Soundness:** 4
**Presentation:** 3
**Contribution:** 1
**Rating:** 8
**Confidence:** 3

**Summary:**

The authors propose to extend RFF with parametric transforms of the Fourier features of shift invariant kernels. A path similar to the seminal work of Gretton et al 2007 is followed, where the proposed $\text{HSIC}_\omega$ statistic is shown to converge in distribution to a Gaussian with the true mean (under the null). Then, linear time estimates of the first two moments are provided, which are needed in vanilla HSIC to analytically compute an approximation to the critical region.

**Strengths:**

This is a work that has the potential to be a seminal paper in the field of conditional independent testing for large scale data.

The major improvement is the ability to incorporate modern learning methods in the well-established HSIC framework, while controlling the typically quadratic complexity.

The claims come with a concentration bound of the proposed statistic, linear time 1st and 2nd moment estimators, a uniform (upper) bound for the convergence of the optimisation and a probabilistic upper bound of Type II error given a lower bound on the dimension, that depends on the dataset.

This is a simple idea but the supporting theory is far from trivial.

**Weaknesses:**

The main weakness of this method is its limitation to translation invariant kernels, which is, however, quite a usual case.

Another weakness is that the dataset must be split in two parts to avoid the selective inference problem.

Finally, as an extension of HSIC, this method most likely also suffers from some hindrances in the latter, for instance that the Null distribution (or, in specific, the necessary quantile) has to be empirically approximated, say using bootstrapping, or a possibly looser bound needs to be used.

Some comments:
* Table 1 needs some more space around it to avoid confusion.
* ln 51: statistics
* $\xi_\omega$ needs to be defined in  Theorem 2, and for completeness also $\sigma_\omega$, $\hat\sigma_\omega$.
Perhaps these could be added in the Appendix as a per-lemma symbol table?

**Questions:**

I had difficulty following the implications of Theorem 2, given its (reasonably) over-packed formulation and I believe that a couple more sentences explaining the involved quantities would make it easier to read.

**Limitations:**

This work is limited to translation invariant kernels which can be expressed as an integral using the spectral measure, as well as the dataset splitting issue.

A brief discussion could improve the proposed work by making these issues explicit.

---

> ### Author Rebuttal · Authors · 2024-08-03
>
> Thank you very much for your positive comments on our work. In what follows, we tried to respond to your concerns, and hope that this feedback is helpful for clear up your concerns.
>
> ***W1***: The main weakness of this method is its limitation to translation invariant kernels, which is, however, quite a usual case.
>
> ***Response to W1***: Translation-invariant is a common and practical assumption, which is primarily used to ensure the characteristic property [1] of the kernel and as a prerequisite for using frequency-domain approximations. However, this assumption can be relaxed in some cases. For example, in the case of deep kernel [2], which may lack translation invariance. [2] demonstrates that the kernel's characteristic is preserved when the neural network meets certain assumptions. In our case (see lines 139-141, section 4.2), the theoretical properties are also guaranteed under similar assumptions about the feature map T (the notion is in line 139 Section 4.1).
>
> Regarding the necessity of this assumption under frequency-domain approximation, our framework (line 139-142 Section 4.1) provides a relaxation. We can use T to handle the non-translation-invariant part, while performing frequency-domain approximations on the translation-invariant part to achieve speedup. Thus, this assumption can be relaxed for specific data scenarios by using our framework in conjunction with a specific model design.
>
> ***W2***: Another weakness is that the dataset must be split in two parts to avoid the selective inference problem.
>
> ***Response to W2***: The splitting strategy has an advantage of being able to address the overfitting issue and effectively control Type I errors, ensuring the validity of the test (section 4.3, lines 207-209).
>
> Currently, alternative approaches are designed for two main scenarios: one involves selecting kernels from a finite/countable set (referred to as the discrete scenario) and the other involves performing kernel parameter searches in a continuous space (referred to as the continuous scenario). For the discrete scenario, some methods [3,4] control Type I errors by applying techniques from the selective inference literature. However, these methods cannot be applied to a continuous scenario due to the uncountable set of kernels involved. To the best of our knowledge, both our scheme and existing methods [2,5] rely on data splitting for the continuous case. Designing methods to control Type I errors in the continuous case without sample splitting is an intriguing problem, which is our future work.
>
> The following references are cited in the our paper.
>
> [1] Sriperumbudur, B. K., Gretton, A., Fukumizu, K., Schölkopf, B., and Lanckriet, G. R. (2010). Hilbert space embeddings and metrics on probability measures. The Journal of Machine Learning Research, 11:1517–1561
>
> [2] Liu, F., Xu, W., Lu, J., Zhang, G., Gretton, A., and Sutherland, D. J. (2020). Learning deep kernels for non-parametric two-sample tests. In: International Conference on Machine Learning (ICML 2020), pages 6316–6326. PMLR.
>
> [3] Kübler, J., Jitkrittum, W., Schölkopf, B., and Muandet, K. (2020). Learning kernel tests without data splitting. Advances in Neural Information Processing Systems, 33:6245–6255.
>
> [4] Schrab, A., Kim, I., Guedj, B., and Gretton, A. (2022). Efficient aggregated kernel tests using incomplete u-statistics. Advances in Neural Information Processing Systems, 35:18793–18807.
>
> [5] Jitkrittum, W., Szabó, Z., and Gretton, A. (2017). An adaptive test of independence with analytic kernel embeddings. In International Conference on Machine Learning, pages 1742–1751. PMLR.
>
> ***W3***: Finally, as an extension of HSIC, this method most likely also suffers from some hindrances in the latter, for instance that the Null distribution (or, in specific, the necessary quantile) has to be empirically approximated, say using bootstrapping, or a possibly looser bound needs to be used.
>
> ***Response to W3***: Yes, most current HSIC variants, including our method, require empirical approximation to determine the threshold. One possible reason is that the HSIC statistic is not normalized, necessitating the computation of certain distributional parameters of the asymptotic distribution under H0, such as the mean and variance. Exploring how to integrate our framework with the development of new statistics to bypass this issue is an interesting issue for future work.
>
> ***Response to the comments for writing***: Thanks for your suggestions, we will fix these issues in the revised version.
>
> ***Questions***:
> I had difficulty following the implications of Theorem 2, given its (reasonably) over-packed formulation and I believe that a couple more sentences explaining the involved quantities would make it easier to read.
>
> ***Response to questions***: Theorem 2 can be divided into two key results. The portion in lines 237-239 addresses the first result, which states that the test power converges to 1, indicating that the test is consistent under the condition $\mathbf{E}_Z\text{HSIC}^{(u)}_{\omega}(Z^{te})>0$. The second part, detailed in lines 239-244, examines the conditions under which this requirement is met, which necessitates sufficient frequency sampling. Overall, with adequate frequency sampling, our test guarantees consistency (lines 245-251). I hope this will make it easier for you to understand the implications of Theorem 2.
>
> Anyway, we will revise Theorem 2 to make it easier to be understood in the revised manuscript.
>
>
> ***Limitations***:
> This work is limited to translation invariant kernels which can be expressed as an integral using the spectral measure, as well as the dataset splitting issue.
> A brief discussion could improve the proposed work by making these issues explicit.
>
> ***Response***：see our response to W1.

---

### Official Review · Reviewer_gka3 · 2024-07-14

**Soundness:** 2
**Presentation:** 2
**Contribution:** 2
**Rating:** 4
**Confidence:** 3

**Summary:**

The paper proposes a novel method of estimating HSIC to determine the independence of two random variables. The original formulation HSIC_b is reformulated and approximated using Monte Carlo integration, HSIC_w, with frequency samples. This reformulation was not derived by the authors but was borrowed from previous literature. The authors’ contribution is the sampling of frequencies proportionally to the inverse Fourier transform of kernel functions.

**Strengths:**

The formulations derived in the paper are mostly reasonable. When the inverse transform of the kernel resembles the equation of well-known density functions, HSIC can be approximated using frequency samples. For that kind of kernels, Gaussian (or Mahalanobis) and Laplace kernels are introduced. For other kernels, it is not clear whether their inverse Fourier transformations can result in the form of well-defined probability density functions. Even for Laplace kernel, domain of w_d is not provided, and its density function for frequency sampling is not well-defined.

Originally, the calculation of a pair of samples could be calculated using a single equation, but it is designed to be calculated using samples. It looks like an unnecessary additional cost, but the calculation of nxn matrix multiplications finally have been converted into the calculation of DxD matrix multiplications as in Eq. 10, and I enjoyed reading the conversion of the equation. The matrix multiplication for the calculation of original HSIC costs O(n^3) with the number of data n, but the proposed method provided an algorithm O(nD^3) with the number of frequency samples D. The calculation should significantly reduce the cost at the expense of the accuracy.

**Weaknesses:**

The proposed estimation is novel, but there are several downsides of the paper.

Many explanations are simply provided without proper self-contained information. First, the purpose of constructing criterion J is unclear. The introduction of $c_\alpha$ is unclear. Second, the definition of Type I error for evaluation is not provided. Type I error is for the false positives. What is the definition of positive decisions in the experiment? What is the test power in the experiment? Those concepts are used without definition.

The authors used neural networks for T_\theta x. Conventional HSIC can also use similar transformation of x. For example, k(T_\theta x, T_\theta x’) can be calculated directly with a learned transformation T_\theta x. The results do not necessarily support the superiority of the calculation in the frequency domain.

In the experiments, I expected to see the consistency of the proposed estimator with large D, which is not provided.

In theory, the convergence is derived for E[HSIC_w], not for HSIC_b.

Minor comments:
In the last line of Algorithm 1, HSIC_b should be HSIC_w.

**Questions:**

Could you provide a detailed description of the objective J, Type I error, and test power used in the paper?

Please explain if we can use k(Tx_1, Tx_2) instead of using the proposed method.

**Limitations:**

The authors presented an interesting algorithm, but there are several notions that are unclear.

---

> ### Author Rebuttal · Authors · 2024-08-03
>
> Thanks for your detailed comments. We’ve made a point-point response to your comments. We would appreciate that you can check our feedback. We hope our feedback can clarify most of your concerns, and we are looking forward to your further questions.
>
> ***W1***: About criterion J , $c_\alpha$, Type I error, and test power.
> ***Q1***: need a detailed description of J and concepts.
>
> ***Response to W1, Q1***:
>
> 1) *About the purpose of constructing criterion J and $c_\alpha$*
>
> **Note that the purpose of constructing J was clearly described in Sec. 4.2 (lines 161-162): “Next, we model the behavior of $\text{HSIC}_\omega(Z)$ to obtain an optimization objective for maximizing the power of the test.” Furthermore, the derivation of J is according to the motivation for constructing J (Section 4.2).** Here, we make a summary of the derivation process.
>
> By the definition of test power, we first model the power theoretically as in Sec. 4.2 (lines 166-172), which have three terms. We then try to obtain an estimate, which is actually the criterion J defined in line 203.
> The criterion J consists of three main terms:
> a) Estimate of the Statistic: the term is an estimate of the test statistic, as given in lines 177-178, Sec. 4.2.
> b) **Threshold $c_\alpha$: This term is used as the threshold for the test to control type I error and is calculated using the two moments of the distribution under the null hypothesis H0. (lines 185-190, Sec. 4.2).**
> c) Variance Estimate under H1: The remaining term is the estimate of the variance of the distribution under the alternative hypothesis H1 as given in line 194, Sec. 4.2.
>
> After defining J, we use it for learning the parameterized kernel as described in Alg. 1 (lines 4-7, Sec. 4.3). This process ensures that the kernel is optimized to enhance the power of the independence test while controlling for Type I error.
>
> 2) *About Type I/II error and test power*
>
> Actually, **Type I/II errors and the test power are ordinary concepts in independence tests, and we describe these concepts in Sec. 2 (lines 78-80)**: “Two types of errors may occur in this procedure. Type I error occurs when $\mathcal{H}_0$ is falsely rejected, while Type II error happens when $\mathcal{H}_0$ is incorrect but not rejected. A good test needs to control Type I error while maximizing the testing power (1-Type II error).” and their evaluation can also be implemented according to the definitions provided.
>
> In experiments, we evaluate Type I/II error and test power, as follows:
>
> **Type II error evaluation**: 1. Generate Dependent Samples: Create pairs X and Y that are dependent. 2. Test Execution: Record the results—*0* if the test return "X is independent with Y", *1* otherwise. 3. Calculate Error Rate: Repeat 100 times, and compute the mean. If 20 out of 100 results is *0*, the Type II error rate is 0.2.
>
> **Test power evaluation**: The test power, therefore, is 1− Type II error rate, which in this case is 0.8.
>
> **Type I error evaluation**: 1. Generate Independent Samples: Create pairs X and Y that are independent. 2. Test Execution: Record the results of the test—*0* if the test return "X is independent with Y", *1* otherwise. 3. Calculate Error Rate: Repeat 100 times. If 5 out of 100 results is *1*, the Type I error rate is 0.05.
>
> ***W2, Q2***: About k(T_\theta x, T_\theta x’).
>
> ***Response to W2, Q2***: Absolutely, the idea you mentioned is NOT acceptable, as it **has the same time and space complexity O(n^2) as the "conventional HSIC". In contrast, our method has linear complexity, due to our calculation in the frequency domain.**
>
> Specifically, after obtaining T_{\theta} through our learning process with the criterion J, our next goal is to use it for computing the statistic for the independence test.
>
> In your idea, by computing k(T_\theta x, T_\theta x’) and estimating it with samples to get the statistic. **This will result in a complexity of O(n^2) in both time and memory, similar to the "conventional HSIC".**
>
> In contrast, **performing calculations in the frequency domain provides significant benefits.** Specifically, it allows us to compute the statistic (as in our Eq. (10)) with a complexity of O(n) in both time and memory. This advantage clearly demonstrates the superiority of our frequency domain approach.
>
> Additionally, we should **point out that your question presupposes the acquisition of learned T. However, accomplishing this step efficiently is also a major contribution of our paper.** Furthermore, our criterion for learning T is designed to straightforwardly model the test power of our statistic, making it a better match compared to your method.
>
> ***W3***: about the consistency.
>
> ***Response to W3***: Actually, **the consistency of the test has already provided in the experiments. In the experiments (Sec. 6.1, Lines 292-293)**, we state that “In addition, as the sample size increases, the test power of LFHSIC-G/M is gradually converging to 1 in both settings, which corroborates the results of Theorem 3.” This observation conforms to the consistency of our test.
>
> To be more detailed, **the consistency of test is defined as “the power of the test tending to 1 as the sample size increases” (Line 235).** The theoretical result of consistency is provided in Theorem 3 (Line 237). This theorem formally concludes that under certain conditions (D is sufficient large), our test can reliably detect dependencies as the number of samples grows, ensuring that the test power approaches 1.
>
> ***W4***: In theory, the convergence is derived for E[HSIC_w], not for HSIC_b.
>
> ***Response to W4***: Actually, in theoretical analysis, we **derived that HSIC_w converges to E[HSIC_w] as in Line 601, and derived that HSIC_w converges to HSIC_b as in Corollary 1 (Line 686)**. Additionally, the relationship between E[HSIC_w] and E[HSIC_b] can be derived since |E[HSIC_w]−E[HSIC_b]|≤E|HSIC_w - HSIC_b|.
>
> ***Response to minor comments***: Thanks for pointing this out, we will fix it in the revised version.

---

> > ### Author Response · Authors · 2024-08-12
> > **Would you please check our rebuttal! Thanks a lot!**
> >
> > Dear reviewer,
> >
> > We submitted our rebuttal to your comments a few days ago. Would you please check our feedback and retrun your further comments?
> >
> > We are looking forward to your futher feedback on any issues of our paper!
> >
> > Thanks  again!

---

### Official Review · Reviewer_X3Av · 2024-07-25

**Soundness:** 3
**Presentation:** 2
**Contribution:** 2
**Rating:** 5
**Confidence:** 3

**Summary:**

The paper presents a novel consistent estimator for HSIC, which is computationally efficient, and tries to maximize the testing power under controlled type-1 error. The idea is to begin with a known relation between HSIC and Fourier-based distance between the relevant characteristic functions. The estimator can be understood as a sample based approximation of this. For standard kernels, it is noted that the distribution from which the samples need to be generated is from a Gaussian. In general, it is assumed that a transformation of variables converts this distribution as a standard Gaussian. It is shown that this estimator has linear computational complexity, for given transformation parameters.

Further, it is proposed to learn the transformation's parameters (which in turn is same as learning the kernel parameters). And the objective for learning is proposed to be maximizing test power under controlled t1error. To this end, using asymptotic convergence arguments, an expression for this objective is analytically arrived at (prop1). It is discussed how to estimate this objective again in linear time (theorem1). Consistency of this objective and the overall estimator are are proved in theorems 2,3.

Simulations on synthetic and benchmarks datasets show the improved test power and computational complexity when compared with existing baselines.

**Strengths:**

1. The idea of sample based estimation of the Fourier-characteristic function based HSIC equivalent seems novel. This is also interesting because it leads to linear time complexity.

**Weaknesses:**

1. It would have been nice if the introduction section had more description of the methodology at an intuitive level. Currently, most of the introduction is motivation, related work and the actual methodology is restricted to one small para. This would give the reader a intuitive understanding and help to understand the later sections easily. Without this it took me some effort to understand what is happening in the overall methodology. (I may be still missing some important points?)

2. The derivations and the main algorithm in section 4.2,4.3 have a very close resemblance with [30] https://proceedings.mlr.press/v238/ren24a/ren24a.pdf . This significantly weakens the contribution and novelty. More importantly, main details in this section could have been cited from [30] or postponed to Appendix. The current presentation seems to give an impression that the sections are entirely new.

3. Since [30] is a close methodology, empirical comparison with it seems to be crucial. However this seems missing.

**Questions:**

few minor comments:

1. notations at places is a bit confusing: e.g., line 121, 146 w_x means different etc.


--- after reading the author responses and other reviews I increase my score.

**Limitations:**

yes

---

> ### Author Rebuttal · Authors · 2024-08-03
>
> Thanks for your comments. We’ve made a point-point response to your comments. We hope that our feedback can clarify most of your concerns or misunderstandings, and we are looking forward to further discussing with you on any issues about our work.
>
> ***W1***: It would have been nice if the introduction section had more description of the methodology at an intuitive level. Currently, most of the introduction is motivation, related work and the actual methodology is restricted to one small para. This would give the reader a intuitive understanding and help to understand the later sections easily. Without this it took me some effort to understand what is happening in the overall methodology. (I may be still missing some important points?)
>
> ***Response to W1***: Thanks for the suggestion. Actually, combining the last two paras of our Introduction section, you can get a deeper and more complete understanding of our work. Anyway, we will add a more detailed introduction to our method in the Introduction section of the revised manuscript.
>
> ***W2***: The derivations and the main algorithm in section 4.2,4.3 have a very close resemblance with [30] https://proceedings.mlr.press/v238/ren24a/ren24a.pdf . This significantly weakens the contribution and novelty. More importantly, main details in this section could have been cited from [30] or postponed to Appendix. The current presentation seems to give an impression that the sections are entirely new.
>
> ***Response to W2***: Our paper and [30] may appear some similarity in presentation and writing style, this is primarily due to their goals are all to maximize the power of the test, a concept also applied to other tasks as in [1]. However, presentation and writing style are not necessarily relevant to contribution and novelty. Actually, there are **significant differences between our work and [30]**.
>
> First of all, the motivations of the two works are different: [30] focuses on solving the kernel learning problem but retains a time and space complexity of O(n^2), thus cannot used in large scale setting. In contrast, **our work aims not only to solve kernel learning but also to ensure that the final test can efficiently handles large-scale data, as outlined in the introduction (lines 44-47).**
>
> Secondly, their methods are different. Our method involves **designing both new statistics and criteria for kernel learning.** For example, while [30] derives results based on the asymptotic distribution of HSIC_b , our work is based on HSIC_w.
>
> Thirdly, our method has a **significant advantage over [30]**: the criteria used in [30] have a complexity of O(n^2) in both time and memory, whereas our criteria for learning are designed to have a complexity of O(n). This is a significant advance! This makes our method can efficiently handling large-scale datasets.
>
> [1] Liu, F., Xu, W., Lu, J., Zhang, G., Gretton, A., and Sutherland, D. J. (2020). Learning deep kernels for non-parametric two-sample tests. In: International Conference on Machine Learning (ICML 2020), pages 6316–6326. PMLR.
>
> ***W3***: Since [30] is a close methodology, empirical comparison with it seems to be crucial. However this seems missing.
>
> ***Response to W3***: As we pointed out above, our work and [30] addressed the same problem, but they are different methods with different contributions. And actually, we HAVE conducted extensive empirical comparison between our method and [30]. Please check Lines 260-262 in Section 6: "Additionally, for the comparative methods [30] relevant to us, due to their high time overhead and inability to handle some evaluation settings, we separately provide a comparison with our method under certain feasible experimental settings. The results are given in the Appendix." .
>
> **Due to space limit, we moved the detailed empirical comparison results to Appendix K2 and K3.** Our conclusion is that "Our test consistently results in a better power-runtime tradeoff at different D settings." (Appendix K2, line 855). The key advantage of our method over [30] is in computational efficiency: our criterion for learning has a time complexity of O(n) compared to their O(n^2), and our method requires O(n) storage versus their O(n^2).
>
> As shown in Appendix K3, their method takes over 200 seconds for one test with a sample size of 5000. Given the need to evaluate methods on larger settings (ISA 10000, 60000) with 100 repeats to determine the rate of type I and type II errors, [30] cannot handle these settings. In fact, when n=60000, the memory required to store the kernel matrix of size 60000x60000 in [30] is impractical for general devices. In contrast, our test can "complete a test within 10 seconds even with 100,000 samples." (Appendix K3, lines 870-890), enabling us to handle large datasets very efficiently.
>
> ***Minor Comments***: notations at places is a bit confusing: e.g., line 121, 146 w_x means different etc.
>
> ***Response***: Thanks for pointing this out, we will add some details to explain in the new version.

---

> > ### Comment · Reviewer_X3Av · 2024-08-10
> >
> > Dear authors, I agree that the motivation is different from [30]. Also, I   appreciate the linear time complexity and the Fourier based idea. However my main complaint is that the derivations in sections 4.2,4.3 more or less carry forward from [30]. Infact, most of the material , it seems, can be cited from [30] and details postponed to appendix. Am I missing something ?

---

> > > ### Author Response · Authors · 2024-08-11
> > >
> > > Dear Reviewer,
> > >
> > > Thank you very much for your prompt feedback on our rebuttal. You asked us whether you missed something. Frankly, but without any intention to offense you, we think that you did miss something --- the most important things about our submission, that are, 1) contribution: we propose a new method for Statistical Independence Testing with $O(n)$ time and space complexities, while the method in [30] has time and space complexities of $O(n^2)$. This is absolutely a significant contribution to the area. 2) Novelty: to the end, we solve the problem from the frequency domain perspective, which is a novel solution, completely different from that of [30].
> > >
> > > Our paper and [30] are all about independence testing by learning kernels, that is, they all try to solve the same problem but with different methods, specifically, we tried to develop a more efficient method with linear complexity. It is very normal that our paper and [30] use similar notation systems, have a similar background introduction or preliminaries, and adopt a look-alike derivation process. In our point of view, the value of a paper depends on its contribution and novelty, not its writing style and derivation process.
> > >
> > > If possible, we would appreciate it if you could spend a little more time to have a more careful comparison between our paper and [30], and we are sure you will see the significant difference between the two works.
> > >
> > > We are looking forward to further discussing with you any issues about our paper. Thanks again!

---

### Decision · Program_Chairs · 2024-09-25

**Decision:**

Accept (poster)

**Comment:**

This paper takes the Fourier perspective on the popular HSIC criteria, leading to a linear time computable novel criteria. All the reviewers agree that the work is quite novel. It will be great if the authors handle the apprehensions raised by the reviewers appropriately in their final version, if the paper is eventually accepted. I feel the linear computation aspect is perhaps what distinguishes the work from the others and needs to be highlighted appropriately.